# BREAKING THE CURSE OF DIMENSIONALITY FOR LINEAR RULES: OPTIMAL PREDICTORS OVER THE ELLIPSOID

## ABSTRACT

In this work, we address the following question: *What minimal structural assumptions are needed to prevent the degradation of statistical learning bounds with increasing dimensionality*? We investigate this question in the classical statistical setting of signal estimation from $n$ independent linear observations $Y_i = X_i^\top \theta + \epsilon_i$. Our focus is on the generalization properties of a broad family of predictors that can be expressed as linear combinations of the training labels, $f(X) = \sum_{i=1}^n l_i(X) Y_i$. This class — commonly referred to as linear prediction rules — encompasses a wide range of popular parametric and non-parametric estimators, including ridge regression, gradient descent, and kernel methods. Our contributions are twofold. First, we derive non-asymptotic upper and lower bounds on the generalization error for this class under the assumption that the Bayes predictor $\theta$ lies in an ellipsoid. Second, we establish a lower bound for the subclass of rotationally invariant linear prediction rules when the Bayes predictor is fixed. Our analysis highlights two fundamental contributions to the risk: (a) a variance-like term that captures the intrinsic dimensionality of the data; (b) the noiseless error, a term that arises specifically in the high-dimensional regime. These findings shed light on the role of structural assumptions in mitigating the curse of dimensionality.

## 1 INTRODUCTION

Coined by Bellman et al. (1957), the *curse of dimensionality* (CoD) refers to the ubiquity of high-dimensional bottlenecks in computer science. A classical manifestation in statistical learning is the minimax lower bound for non-parametric regression: achieving an $\epsilon$ excess risk over the class of Lipschitz functions $f_\star : \mathbb{R}^d \to \mathbb{R}$ requires an exponential sample complexity $n \gtrsim \epsilon^{-\frac{2}{2+d}}$ (Tsybakov, 2008). This impossibility result shows that learning a generic high-dimensional function is intractable in the worst case, thereby highlighting the necessity of structural assumptions on the target class. A canonical example is linear regression, where the exponential dependence on $d$ is replaced by a minimax risk lower bound of order $\sigma^2 d/n$ for $n \geq d$ (Tsybakov, 2003; Mourtada, 2022). In contrast, when $n < d$ the minimax risk diverges: in the worst case, no predictor can recover $\theta_\star \in \mathbb{R}^d$, even in the absence of noise. This illustrates how, in the high-dimensional regime, the noiseless error can be made arbitrarily large within the minimax framework.

Although unusual from the perspective of classical statistics, the regime where the number of parameters exceeds the number of samples has gained renewed attention in modern machine learning, largely motivated by the widespread use of overparametrized neural networks. Strikingly, the minimax rate for linear functions contrasts with recent results on high-dimensional linear models, which show that under probabilistic assumptions on the covariates (e.g. sub-Gaussianity) the typical error in the $n < d$ regime remains bounded (Krogh and Hertz, 1991; Dobriban and Wager, 2018; Aubin et al., 2020; Bartlett et al., 2020; Hastie et al., 2022; Cheng and Montanari, 2024). In particular, in the noiseless setting the error can even decay faster than the classical $n^{-1}$ rate.

The central aim of this paper is to reconcile these two perspectives. Specifically, we demonstrate that restricting the minimax problem to the class of linear prediction rules (including popular algorithms such as ridge regression and gradient-based methods) and target functions drawn from an ellipsoid suffices to establish finite upper and lower bounds that capture the modern high-dimensional

phenomenology. In doing so, we redeem the minimax framework in the overparametrized regime. Our **main contributions** are:

- Proposition 3.1 gives a characterization of the averaged excess risk for the optimal linear prediction rule under prior distribution on best predictor.

- Theorem 4.1 establishes simple non-asymptotic upper bounds — expressed in terms of the degrees of freedom — for noisy tasks, while Theorems 4.5 and 4.7 provide complementary lower bounds on the variance term of the optimal linear rule.

- We analyze the noiseless case in two regimes: (i) Theorem 4.10, when the covariance matrix has heavy tails, and (ii) Theorem 4.15, when the spectrum decays rapidly. In both cases, we derive non-asymptotic lower and upper bounds, which are shown to be optimal in certain examples.

- Finally, Proposition 5.1 completes our study by establishing a lower bound on the excess risk for a fixed target $\theta_\star$.

**Related work —** The classical non-asymptotic lower bound of $\sigma^2 \frac{d}{n}$ was established by Tsybakov (2003) and later refined by Mourtada (2022). Numerous upper bounds have also been studied in the literature, including those for ridge regression (Hsu et al., 2012) and SGD regression (Yao et al., 2007; Bach and Moulines, 2013; Dieuleveut et al., 2017). High-dimensional asymptotics for ridge(less) regression was studied under different assumptions on the covariate distribution by Krogh and Hertz (1991); Dobriban and Wager (2018); Aubin et al. (2020); Wu and Xu (2020); Loureiro et al. (2021); Hastie et al. (2022); Bach (2024). Sharp non-asymptotic results were also derived in Bartlett et al. (2020); Cheng and Montanari (2024); Misiakiewicz and Saeed (2024). In particular, the noiseless setting was shown to yield rates faster than $1/n$ (Berthier et al., 2020; Aubin et al., 2020; Varre et al., 2021). Finally, works considering a prior on $\theta_\star$ include (Dicker, 2016; Richards et al., 2021). Excess risk rates under source and capacity conditions have been widely studied in the kernel ridge regression literature (Caponnetto and De Vito, 2007; Richards et al., 2021; Cui et al., 2021; Defilippis et al., 2024).

**Notations.** For $n \in \mathbb{N}$, we denote $[n] = \{1, \ldots, n\}$. For two symmetric matrices $A, B$, we use $A \preceq B$ to denote that the matrix $B - A$ is a symmetric semidefinite positive matrix, $A^\dagger$ denotes the Moore-Penrose pseudoinverse. We denote by $\lambda_j(A)$ the $j$-th eigenvalue of $A$ in non increasing ordering. We use index $i$ for inputs and the index $j$ for features.

## 2 SETTING

We consider the statistical regression problem of predicting an output random variable $Y \in \mathbb{R}$ from an input random variable $X \in \mathcal{X} = \mathbb{R}^d$ related by a noisy linear model:

$$Y = X^\top \theta_\star + \epsilon, \tag{1}$$

with $\mathbb{E}[\epsilon|X] = 0$ (well-specified) and $\mathbb{E}[\epsilon^2|X] = \sigma^2$. Given $n$ i.i.d. samples $(X_i, Y_i)$ drawn from the model in Equation (1), our focus in this work is to investigate the hypothesis class of *linear predictor rules*

$$\hat{f}(X) = \sum_{i=1}^{n} l_i(X) Y_i, \tag{2}$$

defined by a (potentially random) function $l_i$ that depends on the training covariates $(X_i)_{i\in[n]}$ and a data-independent source of randomness.

*Example* 2.1 (Linear prediction rules). The class of linear prediction rules, also known as *linear smoothers* (Buja et al., 1989), encompasses several examples of interest in the literature, such as:

- **Ridge(less) regression**: The ridge regression prediction rule is a linear rule with

$$l_i(X) = \frac{1}{n} X_i^\top (\hat{\Sigma}_n + \lambda I)^{-1} X, \tag{3}$$

where $\hat{\Sigma} = 1/n \sum_{i\in[n]} X_i X_i^\top$ is the empirical covariance matrix. Furthermore, $l_i(X) = \frac{1}{n} X_i^\top \hat{\Sigma}^\dagger X$, corresponding to the minimal norm interpolator, is also a linear prediction rule.

- **Gradient flow**: The predictor obtained by running gradient flow with learning rate $\eta > 0$ on a linear model $f(X) = \theta_t^\top X$ from $\theta_{t=0} = 0$ for $t$ defines a linear predictor rule with:

$$l_i(X) = \frac{1}{n} X_i^\top (\eta e^{-\eta t \hat{\Sigma}} + \hat{\Sigma}^\dagger) X$$

  More generally, some (S)GD recursion, minimizing $\ell_2$-penalized quadratic risk, can also be written as a linear predictor rule, see Appendix A for a discussion.

- **Nadaraya-Watson estimator**: Let $K(x, x') = \kappa\left(\frac{x-x'}{h}\right)$ denote a rotationally invariant kernel with bandwidth $h > 0$. The Nadaraya-Watson estimator defines a linear predictor rule with

$$l_i(X) = \frac{\kappa\left(X - X_i/h\right)}{\sum_{j \in [n]} \kappa\left(X - X_j/h\right)}$$

- More generally, any of the above methods can be generalized by considering a fixed feature map $\phi(X)$ of the covariates, while remaining a linear prediction rule. This includes methods such as principal component regression, Nyström (Williams and Seeger, 2000; Smola and Schökopf, 2000) and Random features methods (Rahimi and Recht, 2007), among others.

- A classical statistics example which *is not* a linear prediction rule is the LASSO (Tibshirani, 1996).

Our main goal in this work is to provide general statistical guarantees for the performance of this class of predictors, as quantified by the *population risk*

$$R(f) := \mathbb{E}\left[(Y_{n+1} - f(X_{n+1}))^2\right], \tag{4}$$

over the class of measurable functions $f : \mathcal{X} \to \mathbb{R}$. The statistically optimal predictor $f_\star$ minimizing $R$ for the model in Equation (1), known as the *Bayes predictor*, is given by the conditional expectation $f_\star(X) = \mathbb{E}[Y|X] = \theta_\star^\top X$. This question, therefore, boils down to quantifying how well $f_\star$ can be approximated by a linear prediction rule with a finite batch of data, and how close the corresponding risk is to the *Bayes risk* $R(f_\star) = \sigma^2$. Note that since $\hat{f}$ is data-dependent, the corresponding risk $R(\hat{f})$ is random, and hence our focus will be in studying the averaged excess risk

$$\mathcal{E}_{\sigma^2}(f) := \mathbb{E}[R(f)] - R(f_\star), \tag{5}$$

where the expectation is taken over the training dataset.

*Remark* 2.2. In this paper, we focus on results in expectation. While these results can be extended to high-probability guarantees under suitable assumptions, we chose to present them in expectation to maintain clarity—particularly for the lower bounds, which are inherently more difficult to interpret and especially challenging to establish in the high-probability setting.

A popular approach for bounding the performance of statistical methods for linear problems is the *minimax approach*, consisting of looking at the performance of the best predictor under the hardest possible rule

$$\inf_{\hat{f}} \sup_{\theta_\star \in \mathbb{R}^d} \mathcal{E}_{\sigma^2}(\hat{f}), \tag{6}$$

where the infimum is typically taken over the class of all possible predictors (measurable functions of the data). In other words, the minimax risk describes the performance of the best possible algorithm evaluated on the worst-case data. While it provides a powerful tool for deriving bounds on the risk, it suffers from poor scaling with the dimension $d$, a problem known as the *curse of dimensionality*. For instance, as shown by Tsybakov (2003) and Mourtada (2022),

$$\inf_{\hat{f}} \sup_{\theta_\star \in \mathbb{R}^d} \mathcal{E}_{\sigma^2}(\hat{f}) \geq \begin{cases} \sigma^2 \frac{d}{n} & \text{if } d \leq n, \\ +\infty & \text{if } d > n, \end{cases}$$

thus the minimax risk in Equation (6) diverges with $d$ as soon $d > n$. This is because the optimal prediction function is supported on the span of observations. Then selecting $\theta_\star$ divergent norm leads to infinite minimax risk. This will be mitigated by following assumptions.

Therefore, providing statistical guarantees that remain meaningful for high-dimensional predictors requires assuming further structure on the Bayes predictor.

**Mini-averaged risk** It will be useful to define the optimal averaged excess risk, called mini-averaged risk, where the Bayes predictor is sampled according to a distribution $\nu$ supported on $\Theta \subset \mathbb{R}^d$:

$$\bar{\mathcal{E}}(\nu; \sigma^2) := \inf_{\hat{f}} \mathbb{E}_{\theta_\star \sim \nu} \left[ \mathcal{E}_{\sigma^2}(\hat{f}) \right], \tag{7}$$

where, the infimum is taken on linear predictor rule Equation (2).

**Best predictor on an ellipsoid** In order to mitigate the poor dimensional scaling of the minimax risk, we consider the following assumption on the Bayes predictor.

**Assumption 1** (Ellipsoidal Bayes predictor)**.** We assume the Bayes predictor belongs to an ellipsoid

$$\theta_\star \in \Theta_A = \{\theta \in \mathbb{R}^d \quad \text{s.t} \quad \|A\theta\|_2 = 1\} \subset \mathbb{R}^d, \tag{8}$$

for a positive semi-definite symmetric matrix $A \in \mathbb{R}^{d \times d}$.

A natural choice of distribution $\nu_A$ supported on $\Theta_A$ is $A^{-1}\mathcal{U}(\mathbb{S}^{d-1})$. In Examples 2.4 and 2.5, we explain that this distribution corresponds to a high dimensional assumption depending on the ellipsoid described by $A$.

*Remark* 2.3 (Comparison with the minimax approach)*.* Restricting the Bayes predictor to the ellipsoid immediately provides a lower bound to the unconstrained minimax risk. More interestingly, the optimal averaged risk is also a lower bound to the constrained minimax risk:

$$\inf_{\hat{f}} \sup_{\theta_\star \in \mathbb{R}^d} \mathcal{E}_{\sigma^2}(\hat{f}) \geq \inf_{\hat{f}} \sup_{\theta_\star \in \Theta_A} \mathcal{E}_{\sigma^2}(\hat{f}) \geq \inf_{\hat{f}} \mathbb{E}_{\theta_\star \sim \nu_A} \left[ \mathcal{E}_{\sigma^2}(\hat{f}) \right] = \bar{\mathcal{E}}(\nu_A; \sigma^2). \tag{9}$$

However, note that minimizing the averaged risk does not give an optimal algorithm in the worst-case sense, but rather an optimal algorithm in the typical case.

*Example* 2.4 (Explained variance)*.* In the case of linear model (1), the risk associated with the naive predictor $f = 0$ is

$$\mathbb{E}[Y^2] = \|\Sigma^{1/2}\theta_\star\|_2^2 + \sigma^2. \tag{10}$$

Thus, assuming a bounded second moment for $Y$ is equivalent to assuming that $\theta_\star$ lies within an ellipsoid defined by $\|\Sigma^{1/2}\theta_\star\|_2^2 = \rho^2 > 0$. A bounded *explained variance*, i.e., $\|\Sigma^{1/2}\theta_\star\|_2^2$, is often considered a minimal assumption in regression setting. We discuss the limitations of this assumption in Example 4.13.

*Example* 2.5 (Source condition)*.* A well-known example from the kernel literature satisfying Assumption 1 is the *source condition* Caponnetto and De Vito (2007), which can be seen as an extension of the bounded explained variance assumption. Given $r \geq 0$, the source condition is defined by the ellipsoid described by $\|\Sigma^{1/2-r}\theta_\star\|_2 =: \rho_r$. The constant $r$ parametrizes how fast the target decays with respect to the basis of the covariates, and therefore quantifies the difficulty of the task. To study the source condition, we can take $\nu_r$ such that $\Sigma^{1/2-r}\theta_\star \sim \rho_r \mathcal{U}(\mathbb{S}^{d-1})$. For comparison, we fix $\rho_r^2 = d\rho^2/\text{Tr}(\Sigma^{2r})$, in order to have the average explained variance $\mathbb{E}_\nu \|\Sigma^{1/2}\theta_\star\|_2^2 = \rho^2$ independent of $r$. In this case, the covariance matrix of $\theta_\star$ is given by $H_r = \rho^2 \Sigma^{2r-1}/\text{Tr}(\Sigma^{2r})$.

## 3 OPTIMAL AVERAGED RISK AND ALGORITHM

Our first main result concerns a characterization of the optimal averaged risk for Bayes predictors in the ellipsoid. In the following, we denote by $\Sigma = \mathbb{E}[XX^\top]$ (resp. $\hat{\Sigma} = 1/n \sum_{i \in [n]} X_i X_i^\top$) the population (resp. empirical) covariance matrix of the training covariates.

**Proposition 3.1.** *Let $\nu$ denote a distribution supported on $\Theta$, and denote $H := \mathbb{E}_\nu[\theta\theta^\top] \succeq 0$. For $i \in [n+1]$, define the transformed observation $\tilde{X}_i = H^{1/2}X_i$. Then, the optimal averaged excess risk over the class of linear prediction rules is given by ridge regression on the transformed covariates $(\tilde{X}_i)_{i \in [n]}$ and ridge penalty $\lambda = \frac{\sigma^2}{n}$. In other words, the optimal linear prediction rule is*

$$l_i(X_{n+1}) = \begin{cases} \frac{1}{n} \tilde{X}_i^\top (\hat{\Sigma}_H + \lambda I)^{-1} \tilde{X}_{n+1} & \text{if } \sigma^2 > 0 \\ \frac{1}{n} \tilde{X}_i^\top \hat{\Sigma}_H^\dagger \tilde{X}_{n+1} & \text{if } \sigma^2 = 0, \end{cases} \tag{11}$$

*with averaged excess risk*

- *(Variational form)*

$$\bar{\mathcal{E}}(\nu; \sigma^2) = \mathbb{E}\left[\inf_{l \in \mathbb{R}^n} \left\|\sum_{i=1}^n l_i \tilde{X}_i - \tilde{X}_{n+1}\right\|_2^2 + \sigma^2 \sum_{i=1}^n l_i^2\right]. \tag{12}$$

- *(Matrix form)*

$$\bar{\mathcal{E}}(\nu; \sigma^2) = \frac{\sigma^2}{n}\mathbb{E}\left[\mathrm{Tr}(\Sigma_H(\hat{\Sigma}_H + \lambda I)^{-1})\right], \tag{13}$$

*where $\Sigma_H$ (resp. $\hat{\Sigma}_H$) the population (resp. empirical) covariance matrix of transformed observations $(\tilde{X}_i)_{i \in [n]}$.*

*Remark* 3.2. A few remarks on Proposition 3.1 are in order.

(a) The form of the best linear rule is classical in Bayesian literature (Bishop and Nasrabadi, 2006) and is yet used in Dobriban and Wager (2018); Richards et al. (2021) for studied ridge(less) regression with random design.

(b) Proposition 3.1 shows that the optimal averaged excess risk in Equation (7) only depends on the distribution $\nu$ through its second moment $H$. Furthermore, the optimal risk depends only on the distribution of transformed observations $(\tilde{X}_i)_{i \in [n]}$ of population covariance matrix $\Sigma_H = H^{1/2}\Sigma H^{1/2}$. Thus, to simplify the notation and the reading of the results, from now, we will adopt the notation

$$\bar{\mathcal{E}}(\Sigma_H; \sigma^2) := \bar{\mathcal{E}}(\nu; \sigma^2). \tag{14}$$

Note that $\Sigma_H$ contains both information of the covariance structure of $X$ and the signal $\theta_\star$.

(c) In the case of $\theta_\star \sim \nu_A$ distributed on ellipsoid $\Theta_A$ (Assumption 1), we have $H^{1/2} = \frac{A^{-1}}{\sqrt{d}}$ and $\Sigma_H = \frac{1}{d}A^{-1}\Sigma A^{-1}$. Furthermore, we can complete (9), on the constrained minimax risk, by

$$\bar{\mathcal{E}}(\Sigma_H; \sigma^2) \leq \inf_{\hat{f}} \sup_{\theta_\star \in \Theta_A} \mathcal{E}_{\sigma^2}(\hat{f}) \leq \bar{\mathcal{E}}(d\Sigma_H; \sigma^2). \tag{15}$$

The upper-bound is obtained using Proposition 3.1 variational form. Remarks that the constrained minimax risk is upper bounded as soon as $\Theta$ is an ellipsoid.

(d) Both the matrix and variational form of Proposition 3.1 provide useful intuition on the optimal algorithm. The matrix form is useful to obtain either (i) high-dimensional asymptotic equivalents, for instance with random matrix theory tools such as in Dobriban and Wager (2018); Cheng and Montanari (2024); (ii) lower-bounds using trace operator concavity/convexity properties. Similarly, the variational form is useful to derive upper bounds on the optimal averaged error $\bar{\mathcal{E}}$, for instance by choosing an appropriate linear rule $l_i$ for which the expectation in Equation (13) is easy to compute explicitly.

**Degrees of freedom and the noiseless error** For $k \in \{1, 2\}$, define the *k-th degree of freedom* $\mathrm{df}_k(\Sigma; \lambda) = \mathrm{Tr}(\Sigma^k(\Sigma + \lambda I)^{-k})$. The degrees of freedom is a key quantity to understand $\ell_2$ regularization, and appears in a large number of works on ridge and kernel ridge regression (Caponnetto and De Vito, 2007; Bach, 2017; 2024). It can be interpreted as a soft count of the number of eigenvalues of $\Sigma$ which are smaller than $\lambda$, as $\mathrm{df}_1(\Sigma; \lambda) \simeq k$ if the first $k$ eigenvalues of $\Sigma$ are large with respect to $\lambda$. Using Proposition 3.1, a crude lower bound on the optimal risk is given by

$$\bar{\mathcal{E}}(\Sigma_H; \sigma^2) \geq \sigma^2 \frac{\mathrm{df}_1(\Sigma_H; \lambda)}{n}. \tag{16}$$

This lower bound can be compared to the low-dimensional lower bound for least-squares regression $\sigma^2 d/n$, where $\mathrm{df}_1(\Sigma_H; \lambda)$ plays the role of an effective dimension. However, note that in the noiseless case $\sigma^2 = 0$ this lower bound becomes vacuous, while it is well-known from high-dimensional asymptotics that the excess risk can be non-zero even if $\sigma^2 = 0$ (Hastie et al., 2022).

Capturing this behavior requires a finer analysis of the optimal averaged excess risk. Note that the noiseless optimal excess risk $\bar{\mathcal{E}}(\Sigma_H; 0)$ can be seen as a systematic high-dimensional error. Indeed,

since for $\sigma^2 = 0$ a linear prediction rule takes the form

$$\hat{f}(X) = \sum_{i=1}^{n} l_i(X) X_i^\top \theta_\star, \tag{17}$$

the predictor has information on the target $\theta_\star$ only through the low number $n$ of explored directions $l_i(X)$. Consequently, we have $\bar{\mathcal{E}}(\Sigma_H; \sigma^2) \geq \bar{\mathcal{E}}(\Sigma_H; 0)$ — but this lower bound does not capture the impact of the noise.

This discussion motives the following decomposition of the optimal excess risk

$$\bar{\mathcal{E}}(\Sigma_H; \sigma^2) = \bar{\mathcal{E}}(\Sigma_H; 0) + \bar{\mathcal{E}}(\Sigma_H; \sigma^2) - \bar{\mathcal{E}}(\Sigma_H; 0), \tag{18}$$

where the first term $\bar{\mathcal{E}}(\Sigma_H; 0)$ is the noiseless error, equal to the averaged bias of an overparametrized ridgeless regression problem, but lower than the bias of other linear predictor rules. The second term, $\bar{\mathcal{E}}(\Sigma_H; \sigma^2) - \bar{\mathcal{E}}(\Sigma_H; 0)$, can be interpreted as a variance-like term, since $\bar{\mathcal{E}}(\Sigma_H; \sigma^2) - \bar{\mathcal{E}}(\Sigma_H; 0) = 0$ if $\sigma^2 = 0$. However, it is important to stress that this is not the standard variance of the bias-variance decomposition, since it captures part of the bias of the optimal algorithm.

Our goal in the following will be to derive upper- and lower-bounds for each term in this decomposition.

## 4 UPPER- AND LOWER- BOUNDS ON THE OPTIMAL AVERAGED RISK

In this section we derive statistical guarantees for the optimal excess risk in Proposition 3.1. The discussion will treat the noisy and noiseless cases separately, as these will require different technical tools.

### 4.1 NOISY CASE

We start by discussing the noisy case $\sigma^2 > 0$. Consider the following assumption on the covariate distribution:

**Assumption 2.** There exists $L_H > 0$ such that $\mathbb{E}[\|\tilde{X}\|_2^2 \tilde{X}\tilde{X}^\top] \preceq L_H^2 \Sigma_H$.

Assumption 2 assumption is satisfied for bounded data ($\|\tilde{X}\|_2^2 \leq L_H^2$ almost surely). It is also satisfied by unbounded distributions satisfying the following assumption.

**Assumption 3.** We assume that there exist $\kappa \geq 1$ such that $\mathbb{E}[(v^\top X)^4] \leq \kappa(v^\top \Sigma v)^2$.

In that case, Assumption 2 holds with $L_H^2 = \kappa \text{Tr}(\Sigma_H)$. Assumption 3 is satisfied, for example, with $\kappa = 3$ if $X$ is a Gaussian vector. In particular, the strength of this assumption is that the constant $\kappa$ is invariant under linearly transformations of the covariates. These two assumptions are common in the analysis of linear models, and have appeared before for instance in Bach and Moulines (2013).

**General upper bound —** Our first guarantee is an upper bound on the optimal excess risk under Assumption 2 and for a finite number $n$ of inputs.

**Theorem 4.1.** *Under the setting introduced in Section 2 and Assumption 2,*

$$\lambda \text{df}_1(\Sigma_H; \lambda) \leq \bar{\mathcal{E}}(\Sigma_H; \sigma^2) \leq (\lambda + \lambda_0)\text{df}_1(\Sigma_H; \lambda + \lambda_0), \tag{19}$$

*where $\lambda = \sigma^2/n$, $\lambda_0 = L_H^2/n$.*

*Example* 4.2 (Optimal risk on the sphere). Consider Example 2.5 with $r = 1/2$, corresponding to the best algorithm on the sphere with averaged explained variance equal to $\rho^2$. We have $H_{1/2} = \rho^2 I/\text{Tr}(\Sigma)$ and $\Sigma_{H_{1/2}} = \rho^2 \Sigma/\text{Tr}(\Sigma)$. Then the best predictor is the ridge with $\lambda^\star = \frac{\text{Tr}(\Sigma)}{n} \frac{\sigma^2}{\rho^2}$ and, under Assumption 3, the averaged risk is upper-bounded by

$$\bar{\mathcal{E}}(\Sigma_{H_{1/2}}; \sigma^2) \leq \frac{\sigma^2 + \kappa\rho^2}{n} \text{df}_1(\Sigma; \lambda'), \tag{20}$$

with $\lambda' = \frac{\text{Tr}(\Sigma)}{n} \frac{\sigma^2}{\rho^2} + \frac{\kappa}{n} = \lambda^\star + \frac{\kappa}{n}$. Note that this upper bound is meaningful even if $\sigma^2 = 0$. In particular, note that the ridge penalty $\lambda'$ appearing this upper bound is the sum of two terms: the

optimal ridge regularization $\lambda^\star = \frac{\mathrm{Tr}(\Sigma)}{n}\frac{\sigma^2}{\rho^2}$ and an effective regularization $\lambda_0 = \kappa_1/n$ — which is positive even in the noiseless case $\sigma^2 = 0$. This is akin to the effective regularization observed in the asymptotic analysis of ridge regression (Cheng and Montanari, 2024; Misiakiewicz and Saeed, 2024; Defilippis et al., 2024; Bach, 2024). Interestingly, a similar phenomenon also appears in the context of the optimal excess risk in the class of linear prediction rules.

*Example* 4.3 (Source and capacity conditions). Consider Example 2.5 with $r > 0$. Furthermore, we assume that $\lambda_j(\Sigma) = j^{-\alpha}$. If $r\alpha > 1/2$ then

$$\bar{\mathcal{E}}(\Sigma_{H_r}; \sigma^2) \leq C_{\alpha r}\rho^2 \left( \frac{\sigma^2}{n\rho^2} + \frac{\kappa}{n} \right)^{1 - \frac{1}{2\alpha r}}, \tag{21}$$

with $C_{\alpha r}$ that depends only of $\alpha r$. Thus, the rate decreases with $r$ and $\alpha$, which represent, respectively, the complexity learning of the target $\theta_\star$ and the inputs $X$.

*Remark* 4.4 (Infinite dimensional inputs). Theorem 4.1 extends to the setting where $X$ lies in an RKHS. In fact, Assumption 2 can be generalized to Hilbert spaces via operator theory, and the first degree of freedom is defined whenever $\mathrm{Tr}(\Sigma_H) < +\infty$.

**Lower bounds —** Deriving general lower bounds for the optimal excess risk is more challenging. A first step in this direction is to derive a lower bound for the term $\bar{\mathcal{E}}(\Sigma_H; \sigma^2) - \bar{\mathcal{E}}(\Sigma_H; 0)$, which plays a role similar to a variance in our analysis. Considering notation of Theorem 4.1, we have the following result.

**Theorem 4.5.** *Under the setting introduced in Section 2 and Assumption 2:*

$$C_{\sigma, L_H} \frac{\sigma^2}{n} \mathrm{df}_2\left( \Sigma_H; \lambda_{\sigma, L_H} \right) \leq \bar{\mathcal{E}}(\Sigma_H; \sigma^2) - \bar{\mathcal{E}}(\Sigma_H; 0), \tag{22}$$

*with*

- $C_{\sigma, L_H} = 1 - L_H^2/\sigma^2$ *and* $\lambda_{\sigma, L_H} = \lambda + \lambda_0 = (\sigma^2 + L_H^2)/n$ *if* $L_H^2 < \sigma^2$

- $C_{\sigma, L_H} = 1/(1 + L_H^2/\sigma^2)^2$ *and* $\lambda_{\sigma, L_H} = \lambda = \sigma^2/n$ *if* $\|\tilde{X}\|_2^2 \leq L_H$ *almost-surely.*

*Remark* 4.6. Theorem 4.5 provides two cases in which the variance-like term can be lower-bounded by $\sigma^2 d_{\mathrm{eff}}/n$, where the second degree-of-freedom plays the role of the effective dimension. This is natural given the already highlighted similarities with the ridge regression literature. This lower bound is mostly useful in the noisy case, i.e. when the noise variance $\sigma^2$ is not negligeable with respect to the signal strength and covariate variance, quantified here by $L_H$. In particular, $L_H^2/\sigma^2$ can be interpreted as a signal-to-noise ratio.

Theorem 4.5 can be completed by the following result that shows optimality of Theorem 4.1 under the assumptions considered here.

**Theorem 4.7** (Lower bound on supremum). *Let* $\mathcal{P}(\Sigma_H, L_H^2)$ *denote the set of distributions of covariates* $\tilde{X}$ *with covariance matrix* $\Sigma_H$ *satisfying Assumption 2. Then,*

$$(\lambda + \lambda_0)\mathrm{df}_1\left( \Sigma_H; \lambda + \lambda_0 \right) - \lambda_0 \mathrm{df}_1\left( \Sigma_H; \lambda_0 \right) \leq \sup_{\mathbb{P} \in \mathcal{P}(\Sigma_H, L_H^2)} \left\{ \bar{\mathcal{E}}(\Sigma_H; \sigma^2) - \bar{\mathcal{E}}(\Sigma_H; 0) \right\}.$$

*Remark* 4.8. By construction, this is the tightest lower bound with respect to the upper bound in Theorem 4.1. It corresponds to the difference between the noisy and noiseless cases in Equation (19), implying that this upper bound cannot be improved in the large noise regime. For small noise, the upper bound might not be tight. We expect it to be loose as soon as the following upper bound

$$\bar{\mathcal{E}}(\Sigma_H; 0) \leq \lambda_0 \mathrm{df}_1\left( \Sigma_H; \lambda_0 \right),$$

becomes loose. However, we note that the variance-like term is sub-proportional to the noise variance, and therefore in the weak noise regime the contribution from this term is sub-leading.

## 4.2 Noiseless case

In the last section, we saw that we can derive fairly general upper- and lower-bounds for the optimal excess risk over the class of linear predictors which tightness depend on the noise level, and in

particular become loose as the noise variance vanishes. Our goal in this section is to investigate the optimality of the upper bound in Theorem 4.1 in the noiseless case $\sigma^2 = 0$, which is explicitly given by:

$$\bar{\mathcal{E}}(\Sigma_H; 0) \leq \lambda_0 \mathrm{df}_1(\Sigma_H; \lambda_0), \tag{23}$$

with $\lambda_0 = \frac{L_H^2}{n}$. In particular, we recall that under Assumption 3, we have $\lambda_0 = \kappa \frac{\mathrm{Tr}(\Sigma_H)}{n}$. For convenience, we also recall that the average noiseless risk is equal to:

$$\bar{\mathcal{E}}(\Sigma_H; 0) = \mathbb{E}\left[\inf_{l \in \mathbb{R}^n} \left\|\sum_{i=1}^n l_i \tilde{X}_i - \tilde{X}\right\|_2^2\right], \tag{24}$$

which can be rewritten as

$$\bar{\mathcal{E}}(\Sigma_H; 0) = \mathbb{E}\left[\mathrm{Tr}(\Sigma_H(I - P_n))\right], \tag{25}$$

where $P_n$ is the orthogonal projection on the the space spanned by $(\tilde{X}_i)_{i \in [n]}$.

*Remark* 4.9 (Specificity of the noiseless case). A particular property of the noiseless model is that the projection $P_n$ does not depend on the norm of each input $\tilde{X}_i$.

Remark 4.9 motivates the following assumption.

**Assumption 4** (Isotropic latent variable). The latent covariates $Z = \Sigma^{-1/2}X$ satisfy $Z/\|Z\|_2 \sim \mathcal{U}(\mathbb{S}^{d-1})$.

**Implicit noise —** As noted in Theorem 4.1, the term $\lambda_0$ acts as an implicit regularization. Indeed, based on Proposition 3.1, this regularization effect emerges specifically when $\sigma^2 > 0$, since the optimal penalization parameter is given by $\lambda = \sigma^2/n$. In other words, noise induces regularization. This raises the question: how can we explain the presence of the extra term $\lambda_0 > 0$ in the noiseless upper bound? The following theorem shows that this term is not merely an artifact of the analysis, but rather reflects a genuine underlying phenomenon.

**Theorem 4.10.** *Consider the overparametrized case where $d > n + 2$. Then, under the setting introduced in Section 2 and Assumption 4:*

$$\underline{\lambda}_0 \mathrm{df}_1(\Sigma_H; \underline{\lambda}_0) \leq \bar{\mathcal{E}}(\Sigma_H; 0) \leq \bar{\lambda}_0 \mathrm{df}_1(\Sigma_H; \bar{\lambda}_0), \tag{26}$$

*where $\underline{\lambda}_0 = \sigma_0^2/n > 0$, $\bar{\lambda}_0 = {}^{3\mathrm{Tr}(\Sigma_H)}/n$, where $\sigma_0^2$ satisfies, for all $k > n + 2$,*

$$\sigma_0^2 \geq (k-1)(k-n-2)\left(\sum_{j=2}^k \lambda_j(\Sigma_H)^{-1}\right)^{-1}.$$

*Remark* 4.11. Theorem 4.10 can be interpreted as follows:

(a) The upper bound in Theorem 4.10 controls the convergence rate. Intuitively, it corresponds to the contribution of the first degree of freedom and a penalization parameter that scales proportionally to $1/n$.

(b) The parameter $\sigma_0^2$ emerges as the variance of an *implicit noise* in the problem. Indeed, this interpretation is intuitive from the proof, where the leading eigenvectors of $\Sigma_H$ are perturbed due to interactions with the large number of remaining eigenvectors. This is consistent with known upper bounds for linear regression in the overparametrized regime $d > n$, where it was shown that the effects of high-dimensionality can be captured by inflated noise levels (Bartlett et al., 2020; Hastie et al., 2022).

(c) The noise variance $\sigma_0^2$ can be lower-bounded across a broad class of scenarios, including those involving decaying eigenvalue. However, the relevance of the bounds depends on the decay rate of the spectrum. For instance, in the case of geometric decay, the gap between $\underline{\lambda}_0$ and $\bar{\lambda}_0$ can be significant, potentially limiting the tightness of the bound.

*Example* 4.12 (Implicit noise of an isotropic covariance matrix). If $\Sigma = I$ then

$$\sigma_0^2 \geq (d - n - 2) = \left(1 - \frac{n+2}{d}\right)\mathrm{Tr}(\Sigma).$$

*Example* 4.13 (Bounded explained variance). Consider Example 2.5 with $r = 0$. The associated covariance matrix is $\Sigma_{H_0} = \rho^2 I / d$. Theorem 4.10 implies the noiseless error is bounded by

$$\rho^2 \left( 1 - \frac{n+2}{d} \right) \leq \bar{\mathcal{E}}(\Sigma_{H_0}; 0) \leq \rho^2.$$

We observe that the optimal risk suffers from the curse of dimensionality for any $\Sigma \succ 0$, as it converges to the worst-case excess risk $\rho^2$ as the dimension increases. This highlights that a bounded explained variance is not a sufficient assumption in high-dimensional settings.

To complete these examples, we consider the following well-known family of spectra.

**Corollary 4.14.** *Under assumptions of Theorem 4.10 and assume that $\lambda_j = j^{-\alpha}$ (capacity condition) for $\alpha \in (0, 1)$, then*

$$c\bar{\lambda}_0 \mathrm{df}_1 \left( \Sigma_H; \bar{\lambda}_0 \right) \leq \bar{\mathcal{E}}(\Sigma_H; 0) \leq \bar{\lambda}_0 \mathrm{df}_1 \left( \Sigma_H; \bar{\lambda}_0 \right), \tag{27}$$

*with, $c = (1 - \frac{n+2}{d})\frac{(1+\alpha)(1-\alpha)}{12}$ if $\alpha \in (0, 1)$.*

In conclusion, Theorem 4.10 provides optimal bounds (up to a constant) when the spectrum of $\Sigma_H$ decays slowly than $1/j$. For stronger decay Theorem 4.10 is not optimal, but the following theorem can complete this case.

**Theorem 4.15.** *Let $R_k := \sum_{j>k} \lambda_j(\Sigma_H)$. Under assumptions of Theorem 4.10, we have*

$$R_n \leq \bar{\mathcal{E}}(\Sigma_H; 0) \leq \min_{k < n-1} \frac{n-1}{n-k-1} R_k.$$

By choosing different values of $k$, we can obtain various upper bounds. For example, setting $k = n/2$ yields $R_n \leq \bar{\mathcal{E}}(\Sigma_H; 0) \leq 4 R_{n/2}$. The advantage of this bound is that it allows us to exploit the faster decay of the spectrum. In particular, in the context of Example 4.3, the eigenvalues satisfy $\lambda_j(\Sigma_H) \propto j^{-2\alpha r}$ when $2\alpha r > 1$. In the limit $d \to \infty$, we obtain $c_{\alpha r} \rho^2 n^{1-2\alpha r} \leq \bar{\mathcal{E}}(\Sigma_H; 0) \leq C_{\alpha r} \rho^2 n^{1-2\alpha r}$. Hence, the convergence rate is always better than in the noisy case, surpassing $1/n$ when $\alpha r > 1$.

# 5  LOWER BOUND FOR A FIXED TARGET $\theta_\star$

So far, all our results have been derived under the assumption that the target predictor is randomly drawn from the ellipsoid. In this section, we discuss a lower bound result, which exchanges this assumption for the rotationally invariant property:

**Assumption 5.** For any orthogonal matrix $O$, $l_i(X, (X_i)_{i \in [d]}) = l_i(OX, (OX_i)_{i \in [d]})$ almost surely.

Note that all algorithms described in Example 2.1 (excepted LASSO) satisfy this assumption. Following an idea of Richards et al. (2021), we can show that, for any linear rule $\hat{f}$ satisfying Assumption 5, and for a fixed $\theta_\star \in \mathbb{R}^d$, we have $\mathcal{E}_\sigma(\hat{f}) = \mathbb{E}_{\theta_\star \sim \nu} \mathcal{E}_\sigma(\hat{f})$, where $\nu$ is a distribution on $\mathbb{R}^d$ with covariance $H_{\theta^\star} := \sum_{j \in [d]} (v_j^\top \theta_\star)^2 v_j v_j^\top$ where $v_j$ are the eigen-directions of $\Sigma$. Thus, from our results in previous sections, we can show the following proposition.

**Proposition 5.1.** *Under setting of Section 2, Assumption 5, and assuming that $(v_j^\top X)_{j \in [d]}$ have symmetric and independent components, we have*

$$\mathcal{E}_{\sigma^2}(\hat{f}) \geq \bar{\mathcal{E}}(\Sigma_{\theta_\star}; \sigma^2), \tag{28}$$

*where $\Sigma_{\theta_\star} = \sum_{j \in [d]} \lambda_j(\Sigma)(v_j^\top \theta_\star)^2 v_j v_j^\top$.*

*Remark* 5.2. Proposition 5.1 can be interpreted as follows:

(a) The lower bounds in this paper can be used to bound below the excess risk of a specific linear learning rule for a given $\theta_\star$. In particular, thanks to Theorem 4.1, we have

$$\frac{\sigma^2}{n} \mathrm{df}_1(\Sigma_{\theta_\star}; \sigma^2/n) \leq \bar{\mathcal{E}}(\Sigma_{\theta_\star}; \sigma^2) \leq \mathcal{E}_{\sigma^2}(\hat{f}), \tag{29}$$

in the large-noise regime. Moreover, using the decomposition $\bar{\mathcal{E}}(\Sigma_{\theta_\star}; \sigma^2) = \bar{\mathcal{E}}(\Sigma_{\theta_\star}; 0) + \bar{\mathcal{E}}(\Sigma_{\theta_\star}; \sigma^2) - \bar{\mathcal{E}}(\Sigma_{\theta_\star}; 0)$, we can combine the results from Theorems 4.5 and 4.10 to obtain more refined lower bounds.

(b) The lower bound highlights that, to avoid the curse of dimensionality, the optimal predictor $\theta_\star$ must be well aligned with the top eigenvectors of $\Sigma$. For example, if we take $(v_j^\top \theta_\star)^2 = 1/\lambda_j(\Sigma)$, then $\Sigma_{\theta_\star} = I_d$. Applying Theorem 4.10, we obtain $\mathcal{E}_0(\hat{f}) \geq \|\Sigma^{1/2}\theta_\star\|_2^2 \left(1 - \frac{n+2}{d}\right)$. Since $\|\Sigma^{1/2}\theta_\star\|_2^2$ corresponds to the explained variance, this result shows that the predictor is adversely affected by the high dimensionality. In conclusion, assumptions about $\theta_\star$ such as those in Example 2.5, with $r > 0$, are necessary in high-dimensional settings.

## 6 CONCLUSION

This paper establishes that the optimal risk within the class of linear prediction rules can be decomposed into two components. The first is a variance-like term, $\bar{\mathcal{E}}(\Sigma_H; \sigma^2) - \bar{\mathcal{E}}(\Sigma_H; 0)$, which admits a representation in terms of the degrees of freedom. In particular, we show that the lower bound, which depends on the second degree of freedom, takes the form $\sigma^2 d_{\text{eff}}/n$. The second component is the noiseless error $\bar{\mathcal{E}}(\Sigma_H; 0)$, whose decay is governed by the spectral decay of the covariance matrix $\Sigma_H$. For heavy-tailed covariance structures, the noiseless error can be expressed in terms of the first degree of freedom as $\sigma_0^2 d_{\text{eff}}/n$, where $\sigma_0$ accounts for the effective noise generated by the high-dimensional setting. Moreover, when the eigenvalues decay faster than $1/j$, the noiseless error decreases at a rate faster than $1/n$, indicating that the classical rate $d_{\text{eff}}/n$ overestimates the true risk.

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

## A    LINEAR LEARNING RULE

**Proposition A.1** (Linear combination). *If $f$ and $g$ are linear predictor rules then, $\alpha f + \beta g$, where $\alpha$ and $\beta$ are functions of , is a linear prediction rule.*

*Proof.* Writing $f(X) = \sum_{i=1}^{n} l_i^{(f)}(X)Y_i$ and $g(X) = \sum_{i=1}^{n} l_i^{(g)}(X)Y_i$, we proof the result considering $l_i^{(\alpha f + \beta g)} = \alpha l_i^{(f)} + \beta l_i^{(g)}$. $\square$

**Proposition A.2** (Recursion scheme). *All method based on a recursion, starting from $\theta_0 = 0$, of the form,*

$$\theta_t = M_t \theta_{t-1} + \gamma_t Y_{i(t)}, \tag{30}$$

*where $i(t) \in [n], M_t \in \mathbb{R}^{d \times d}$ and $\gamma_t \in \mathbb{R}^d$ are independent of $(Y_i)_{i \in [n]}$ given $(X_i)_{i \in [n]}$, are linear predictor rules.*

*Proof.* We denote by $l^{(t)}$ the linear predictor rule at time $t$.

- $\theta_0$ is linear in $(Y_i)_{i \in [n]}$.

- If $\theta_{t-1}$ is linear in $(Y_i)_{i \in [n]}$ then $\theta_{t-1} = \sum_{i=1}^{n} W_i^{(t-1)} Y_i$ where $W_i^{(t)}$ depends only on $X_i$. Then

$$\theta_t = \sum_{i=1}^{n} M_t W_i^{(t-1)} Y_i + \gamma_t Y_{i(t)}. \tag{31}$$

Then $\theta_t$ is linear in $(Y_i)_{i \in [n]}$.

We conclude using $l_i^{(t)}(X) = X^\top W_i^{(t)}$. $\square$

This shows that any (S)GD method based on minimization of empirical risk, with or without $\ell_2$ penalization, and with or without averaging are linear predictor rules.

## B    PROOF OF SECTION 3

### B.1    PROOF OF PROPOSITION 3.1

**Lemma B.1** (Bias-variance decomposition). *Under setting of Section 2,*

$$\mathbb{E}[(Y - f(X))^2 | X, (X_i)] = \sigma^2 + \left( \left( X - \sum_{i=1}^{n} l_i(X) X_i \right)^\top \theta_\star \right)^2 + \sigma^2 \sum_{i=1}^{n} l_i(X)^2.$$

*Proof.* Starting from $f(X) = \sum l_i(X) Y_i$ and $Y_i = X_i^\top \theta_\star + \epsilon_i$, we have

$$Y - f(X) = \epsilon + X^\top \theta_\star - \sum l_i(X) X_i^\top \theta_\star - \sum l_i(X) Y_i$$

$$= \epsilon + \left( \sum l_i(X) X_i - X \right)^\top \theta_\star - \sum l_i(X) \epsilon_i.$$

Integrating $(Y - f(X))^2$ over $\epsilon, \epsilon_i$ concludes the proof. $\qquad\square$

Thus, we have

$$\mathcal{E}_{\sigma^2}(f) = \mathbb{E}\left[ \left( \left( X - \sum_{i=1}^n l_i(X) X_i \right)^\top \theta_\star \right)^2 + \sigma^2 \sum_{i=1}^n l_i(X)^2 \right]. \tag{32}$$

Integrating this decomposition on $\theta$ and using the Fubini theorem leads to the average excess risk:

$$\mathbb{E}_\nu \mathcal{E}_{\sigma^2}(f) = \mathbb{E}\left[ \left\| X - \sum_{i=1}^n l_i(X) X_i \right\|_H^2 + \sigma^2 \sum_{i=1}^n l_i(X)^2 \right], \tag{33}$$

with $H = \mathbb{E}\theta_\star \theta_\star^\top$. Alternatively, considering the transformed inputs $\tilde{X}_i = H^{1/2} X_i$, we have

$$\mathbb{E}_\nu \mathcal{E}_{\sigma^2}(f) = \mathbb{E}\left[ \left\| \tilde{X} - \sum_{i=1}^n l_i(X) \tilde{X}_i \right\|_2^2 + \sigma^2 \sum_{i=1}^n l_i(X)^2 \right], \tag{34}$$

Thus, the linear rule that minimizes the average excess risk is given by the function $l_i$ that minimizes the integrand $\left\| \sum l_i \tilde{X}_i - \tilde{X} \right\|_2^2 + \sigma^2 \sum_{i=1}^n l_i^2$ (this function will be computed later). Then we obtain the variational form:

$$\bar{\mathcal{E}}(\nu; \sigma^2) = \mathbb{E}\left[ \inf_{l \in \mathbb{R}^n} \left\| \sum_{i=1}^n l_i \tilde{X}_i - \tilde{X}_{n+1} \right\|_2^2 + \sigma^2 \sum_{i=1}^n l_i^2 \right]. \tag{35}$$

For the matrix form, the idea is to consider $l_\star$ the minimizer of $\phi(l) = \left\| \sum l_i \tilde{X}_i - \tilde{X} \right\|_2^2 + \sigma^2 \sum_{i=1}^n l_i^2$. Considering $\mathbf{Z} = (\tilde{X}_1, \ldots, \tilde{X}_d)$ the $\mathbb{R}^{d \times n}$ matrix, we have $\phi(l) = \left[ \| Z - \mathbf{Z}l \|_2^2 + \sigma^2 \| l \|_2^2 \right]$. We use Lemma H.1, to obtain $l_\star = (\mathbf{Z}^\top \mathbf{Z} + \sigma^2 I_n) \mathbf{Z}^\top Z$ and

$$\phi(l_\star) = \sigma^2 \mathrm{Tr}(ZZ^\top (\mathbf{Z}\mathbf{Z}^\top + \sigma^2 I_d)^{-1})$$

$$= \frac{\sigma^2}{n} \mathrm{Tr}\left( \tilde{X}\tilde{X}^\top \left( \hat{\Sigma}_H + \frac{\sigma^2}{n} I \right)^{-1} \right).$$

Then,

$$\bar{\mathcal{E}}(\nu; \sigma^2) = \mathbb{E}\phi(l_\star)$$

$$= \frac{\sigma^2}{n} \mathbb{E}_{X_1, \ldots, X_n} \mathbb{E}_X \mathrm{Tr}\left( \tilde{X}\tilde{X}^\top \left( \hat{\Sigma}_H + \frac{\sigma^2}{n} I \right)^{-1} \right)$$

$$= \frac{\sigma^2}{n} \mathbb{E}\mathrm{Tr}\left( \Sigma_H \left( \hat{\Sigma}_H + \frac{\sigma^2}{n} I \right)^{-1} \right).$$

## B.2 EXAMPLES OF DISTRIBUTION $\nu$

- Uniform distribution on the sphere: If $\theta \sim \mathcal{U}(\mathbb{S}^{d-1})$ using Lemma G.3, we have $\mathbb{E}\theta\theta^\top = \frac{I}{d}$.

- Distribution on ellipsoid described by $\|A\theta_\star\| = 1$: if $A\theta_\star \sim \mathcal{U}(\mathbb{S}^{d-1})$ thus $\theta_\star = A^{-1}\theta$. In consequence, $H = \mathbb{E}\theta_\star\theta_\star^\top = A^{-1}\mathbb{E}\theta\theta^\top A^{-1} = \frac{A^{-2}}{d}$.

- Distribution on source condition $\|\Sigma^{1/2-r}\theta_\star\| = \rho_r$: This corresponds to the previous case with $A = \Sigma^{1/2-r}/\rho_r$. In consequence, $H_r = \rho_r^2 \Sigma^{2r-1}/d$. Not that the average explained variance is $\mathbb{E}\|\Sigma^{1/2}\theta_\star\|_2^2 = \rho_r^2 \mathbb{E}\|\Sigma^{1/2-1/2+r}\theta\|_2^2 = \rho_r \text{Tr}(\Sigma^{2r})/d$. Thus, setting $\rho_r^2 = d\rho^2/\text{Tr}(\Sigma^{2r})$ leads to the same average explained variance over $r \geq 0$.

# C  PROOF OF SECTION 4.1

## C.1  UPPER BOUND OF THEOREM 4.1

*Proof.* The idea is to use variational form of Proposition 3.1 with $l_i(\tilde{X}) = \frac{1}{n}\tilde{X}_i^\top(\Sigma_H + \lambda I)^{-1}\tilde{X}$, with $\lambda > 0$ chosen later. We have

$$\bar{\mathcal{E}}(\Sigma_H; \sigma^2) \leq \mathbb{E}\left[\left\|\tilde{X} - \sum_{i=1}^n l_i(\tilde{X})\tilde{X}_i\right\|_H^2 + \sigma^2\sum_{i=1}^n l_i(\tilde{X})^2\right]. \tag{36}$$

**Step 1 Bias:** We have

$$\sum l_i(\tilde{X})\tilde{X}_i = \frac{1}{n}\sum \tilde{X}_i\tilde{X}_i^\top(\Sigma_H + \lambda I)^{-1}\tilde{X}$$
$$= \hat{\Sigma}_H(\Sigma_H + \lambda I)^{-1}\tilde{X}.$$

Then,

$$\mathbb{E}\left[\left\|\sum l_i(\tilde{X})\tilde{X}_i - \tilde{X}\right\|_2^2\right] = \mathbb{E}\left[\left\|(\hat{\Sigma}_H(\Sigma_H + \lambda)^{-1} - I)\tilde{X}\right\|_2^2\right]$$

$$= \mathbb{E}\left[\left\|(\Sigma_H - \hat{\Sigma}_H + \lambda I)(\Sigma_H + \lambda)^{-1}X\right\|_2^2\right]$$

$$= \mathbb{E}\text{Tr}((\Sigma_H - \hat{\Sigma}_H + \lambda I)(\Sigma_H + \lambda)^{-1}\Sigma_H(\Sigma_H + \lambda)^{-1}(\Sigma_H - \hat{\Sigma}_H + \lambda I))$$

$$= \mathbb{E}\text{Tr}((\Sigma_H + \lambda)^{-2}\Sigma_H(\Sigma_H - \hat{\Sigma}_H + \lambda I)^2)$$

$$= \lambda^2\text{Tr}((\Sigma_H + \lambda)^{-2}\Sigma_H) + \text{Tr}((\Sigma + \lambda)^{-2}\Sigma_H\mathbb{E}[(\Sigma_H - \hat{\Sigma}_H)^2]),$$

using $\mathbb{E}\hat{\Sigma} = \Sigma$. Furthermore,

$$\mathbb{E}[(\Sigma_H - \hat{\Sigma}_H)^2] = \frac{1}{n}\mathbb{E}[(\tilde{X}_1\tilde{X}_1^\top - \Sigma_H)^2]$$

$$= \frac{1}{n}\left(\mathbb{E}[(\tilde{X}_1\tilde{X}_1^\top)^2] - \Sigma_H\right)$$

$$= \frac{1}{n}\left(\mathbb{E}[\|\tilde{X}_1\|_2^2\tilde{X}_1\tilde{X}_1^\top] - \Sigma_H\right)$$

$$\preceq \frac{1}{n}\left(L_H^2\Sigma_H - \Sigma_H\right) \qquad \text{(using Assumption 2.)}$$

Thus, the bias term is bounded by

$$\lambda^2\text{Tr}((\Sigma_H + \lambda)^{-2}\Sigma_H) + \frac{L_H^2}{n}\text{Tr}((\Sigma_H + \lambda)^{-2}\Sigma_H^2).$$

**Step 2:** The variance is given by

$$\sigma^2\mathbb{E}\sum_{i=1}^n l_i(\tilde{X})^2 = \frac{\sigma^2}{n^2}\mathbb{E}\sum_{i=1}^n \tilde{X}^\top(\Sigma_H + \lambda)^{-1}\tilde{X}_i\tilde{X}_i^\top(\Sigma_H + \lambda)^{-1}\tilde{X}$$

$$= \frac{\sigma^2}{n}\text{Tr}((\Sigma_H + \lambda)^{-2}\Sigma_H^2)$$

$$= \frac{\sigma^2}{n}\text{Tr}((\Sigma_H + \lambda)^{-2}\Sigma_H(\Sigma_H + \lambda - \lambda))$$

$$= \frac{\sigma^2}{n}\text{Tr}((\Sigma_H + \lambda)^{-1}\Sigma_H) - \lambda\frac{\sigma^2}{n}\text{Tr}((\Sigma_H + \lambda)^{-2}\Sigma_H).$$

**Step 3:** Putting terms together, $\bar{\mathcal{E}}(\Sigma_H; \sigma^2)$ is upper-bound by

$$\frac{\sigma^2}{n}\mathrm{Tr}((\Sigma_H + \lambda)^{-1}\Sigma_H) + \left(\lambda^2 - \lambda\frac{\sigma^2}{n}\right)\mathrm{Tr}((\Sigma_H + \lambda)^{-2}\Sigma_H) + \frac{L_H^2}{n}\mathrm{Tr}((\Sigma_H + \lambda)^{-2}\Sigma_H^2).$$

Then choosing $\lambda = \frac{\sigma^2}{n} + \frac{L_H}{n}$ leads to $\lambda^2 - \lambda\frac{\sigma^2}{n} = \lambda\frac{L_H}{n}$ and

$$\left(\lambda^2 - \lambda\frac{\sigma^2}{n}\right)\mathrm{Tr}((\Sigma_H + \lambda)^{-2}\Sigma_H) + \frac{L_H^2}{n}\mathrm{Tr}((\Sigma_H + \lambda)^{-2}\Sigma_H^2)$$

$$= \frac{L_H^2}{n}\mathrm{Tr}(\lambda(\Sigma_H + \lambda)^{-2}\Sigma_H + (\Sigma_H + \lambda)^{-2}\Sigma_H^2)$$

$$= \frac{L_H^2}{n}\mathrm{Tr}((\Sigma_H + \lambda)^{-1}\Sigma_H).$$

Finally, we obtain

$$\bar{\mathcal{E}}(\Sigma_H; \sigma^2) \leq \lambda\mathrm{Tr}((\Sigma_H + \lambda)^{-1}\Sigma_H), \tag{37}$$

with $\lambda = \frac{\sigma^2}{n} + \frac{L_H}{n}$. $\qquad\square$

### C.2    LOWER BOUND OF THEOREM 4.1

*Proof.* Using Proposition 3.1 matrix form,

$$\bar{\mathcal{E}}(\Sigma_H; \sigma^2) = \frac{\sigma^2}{n}\mathbb{E}\mathrm{Tr}(\Sigma_H(\hat{\Sigma}_H + (\sigma^2/n)I)^{-1}).$$

Using operator convexity of the inverse (Proposition G.2), we have

$$\bar{\mathcal{E}}(\Sigma_H; \sigma^2) \geq \frac{\sigma^2}{n}\mathrm{Tr}(\Sigma_H(\mathbb{E}\hat{\Sigma}_H + (\sigma^2/n)I)^{-1}) = \mathrm{Tr}(\Sigma_H(\Sigma_H + (\sigma^2/n)I)^{-1}).$$

$\qquad\square$

### C.3    PROOF OF THEOREM 4.5

*Proof of Theorem 4.5.* The first lower bound is just an application of Theorem 4.7. In the follow, we focus on the bounded case with $\|\tilde{X}\| \leq L_H$.

$$\bar{\mathcal{E}}(\Sigma_H; \sigma^2) - \bar{\mathcal{E}}(\Sigma_H; 0) = \frac{\sigma^2}{n}\mathbb{E}\mathrm{Tr}(\Sigma_H(\hat{\Sigma}_H + (\sigma^2/n)I)^{-1}) - \mathbb{E}\mathrm{Tr}(\Sigma_H(I - P))$$

$$= \frac{\sigma^2}{n}\mathbb{E}\mathrm{Tr}(\Sigma_H P(\hat{\Sigma}_H + (\sigma^2/n)I)^{-1}),$$

where $P$ is the orthogonal projection on $\tilde{X}_i$.

$$\bar{\mathcal{E}}(\Sigma_H; \sigma^2) - \bar{\mathcal{E}}(\Sigma_H; 0) = \frac{\sigma^2}{n}\mathbb{E}\mathrm{Tr}(\Sigma_H P(\hat{\Sigma}_H + (\sigma^2/n)I)(\hat{\Sigma}_H + (\sigma^2/n)I)^{-2})$$

$$\geq \frac{\sigma^2}{n}\mathbb{E}\mathrm{Tr}(\Sigma_H\hat{\Sigma}_H(\hat{\Sigma}_H + (\sigma^2/n)I)^{-2})$$

$$=: \frac{\sigma^2}{n}V,$$

because $P\hat{\Sigma}_H = \hat{\Sigma}_H$.

Denoting by $S_n = \sum_{i\in[n]}\tilde{X}_i\tilde{X}_i^\top$, by exchangeability,

$$V = \frac{1}{n}\mathbb{E}[\mathrm{Tr}(\Sigma_H(\hat{\Sigma}_H + \lambda I)^{-1}\tilde{X}_n\tilde{X}_n(\hat{\Sigma}_H + \lambda I)^{-1})]$$

$$= \mathbb{E}[\mathrm{Tr}(\Sigma_H(S_n + n\lambda I)^{-1}\tilde{X}_n\tilde{X}_n(S_n + n\lambda I)^{-1})].$$

Using Sherman-Morrison identity,

$$(S_n + n\lambda I)^{-1}\tilde{X}_n\tilde{X}_n(S_n + n\lambda I)^{-1}$$
$$= \frac{1}{(1 + \|\tilde{X}_n\|^2_{(S_{n-1}+n\lambda I)^{-1}})^2}(S_{n-1} + n\lambda I)^{-1}X_nX_n(S_{n-1} + n\lambda I)^{-1}.$$

Furthermore, $\|\tilde{X}_n\|^2_{(S_{n-1}+n\lambda I)^{-1}} \leq L_H^2/\lambda$. Then, using that $x \longmapsto xM$ increases in $a > 0$ as soon as $M \succeq 0$,

$$\mathbb{E}[(S_{n-1} + n\lambda I)^{-1}\tilde{X}_n\tilde{X}_n(S_{n-1} + n\lambda I)^{-1})|S_{n-1}]$$
$$\succeq \frac{1}{(1 + \frac{L_H^2 n}{\lambda})^2}(S_{n-1} + n\lambda I)^{-1}\Sigma_H(S_{n-1} + n\lambda I)^{-1}).$$

Using convexity of $A \longmapsto ABA$ where $B$ is invertible (Proposition G.2),

$$\mathbb{E}[(S_{n-1} + n\lambda I)^{-1}\tilde{X}_n\tilde{X}_n(S_{n-1} + n\lambda I)^{-1})]$$
$$\succeq \frac{1}{(1 + \frac{L_H^2 n}{\lambda})^2}\mathbb{E}((S_{n-1} + n\lambda I)^{-1})\Sigma_H\mathbb{E}((S_{n-1} + n\lambda I)^{-1}).$$

Using $A := \mathbb{E}((S_{n-1} + n\lambda I)^{-1}) \succeq (\frac{n-1}{n}\Sigma_H + n\lambda I)^{-1} =: B$, we have

$$\mathbb{E}[\text{Tr}(\Sigma(S_{n-1} + n\lambda I)^{-1}\tilde{X}_n\tilde{X}_n(S_{n-1} + n\lambda I)^{-1})] \geq \frac{1}{(1 + \frac{L_H^2 n}{\lambda})^2}\text{Tr}(\Sigma_H A\Sigma_H A)$$
$$\geq \frac{1}{(1 + \frac{L_H^2 n}{\lambda})^2}\text{Tr}(\Sigma_H A\Sigma_H B)$$
$$= \frac{1}{(1 + \frac{L_H^2 n}{\lambda})^2}\text{Tr}(\Sigma_H B\Sigma_H A)$$
$$\geq \frac{1}{(1 + \frac{L_H^2 n}{\lambda})^2}\text{Tr}(\Sigma_H B\Sigma_H B),$$

using that $\Sigma_H A\Sigma_H, \Sigma_H B\Sigma_H \succ 0$. Then,

$$V \geq \frac{1}{(1 + \frac{L_H^2 n}{\lambda})^2}\text{Tr}(\Sigma_H^2((n-1)/n\Sigma_H + n\lambda I)^{-2})$$
$$\geq \frac{1}{(1 + \frac{L_H^2 n}{\lambda})^2}\text{Tr}(\Sigma_H^2(\Sigma_H + n\lambda I)^{-2})$$
$$= \frac{1}{(1 + \frac{L_H^2 n}{\lambda})^2}\text{df}_2(\Sigma_H; \lambda).$$

$\square$

## C.4 LOWER BOUND

We denote by
$$\phi(\lambda) := \lambda\mathbb{E}\text{Tr}(\Sigma(\hat{\Sigma} + \lambda I)^{-1}).$$

We can differentiate over expectancy as soon as $\lambda > 0$.
$$\phi'(\lambda) = \mathbb{E}\text{Tr}(\Sigma(\hat{\Sigma} + \lambda I - \lambda I)(\hat{\Sigma} + \lambda I)^{-2}) = \mathbb{E}\text{Tr}(\Sigma\hat{\Sigma}(\hat{\Sigma} + \lambda I)^{-2}).$$

The idea of the proof of Theorem 4.7 is to lower bound $\phi'$ and then integrate.

*Proof of Theorem 4.7.* We consider the specific distribution satisfying Assumption 2. We consider that $X$ as a discrete distribution along eigenvector of $\Sigma_H = \sum \lambda_j v_j v_j^\top$. More precisely, we choose

$$\mathbb{P}(X = L_H v_j) = \frac{\lambda_j}{\mathrm{Tr}(\Sigma_H)}. \tag{38}$$

Thus, $\hat{\Sigma}_H = L_H \sum_{j \in [d]} N_j u_j u_j^\top$, where $N_j = \sum_{i \in [n]} \mathbf{1}_{X_i = L_H v_j}$ is a binomial distribution. Denoting by, $B_{ij} = \mathbf{1}_{X_i = L_H v_j}$, we have

$$
\begin{aligned}
\mathbb{E}\phi'(\lambda) &= n\mathbb{E} \sum_{j \in [d]} \frac{\lambda_j L_H N_j}{(L_H N_j + n\lambda)^2} \\
&= n \sum_{j \in [d]} \sum_{i \in [n]} \mathbb{E} \frac{\lambda_j L_H B_{ij}}{(L_H N_j + n\lambda)^2} \\
&\geq n \sum_{j \in [d]} \sum_{i \in [n]} \mathbb{E} \frac{\lambda_j L_H B_{ij}}{(L_H \sum_{k \neq j} B_{kj} + L_H + n\lambda)^2} \\
&\geq n \sum_{j \in [d]} \sum_{i \in [n]} \mathbb{E} \frac{\lambda_j^2}{((n-1)\lambda_j + L_H + n\lambda)^2} \qquad \text{(using Jensen inequality)} \\
&= \sum_{j \in [d]} \frac{\lambda_j^2}{(((n-1)/n)\lambda_j + (1/n)L_H + \lambda)^2} \\
&\geq \mathrm{df}_2(\Sigma_H; \lambda + L_H/n).
\end{aligned}
$$

The lower bound is obtained by integration. $\qquad\square$

# D  PROOF OF SECTION 4.2

## D.1  REDUCTION TO THE GAUSSIAN CASE

The projection $P_n$ does not depend of the norm of $\tilde{X}_i = \Sigma_H^{1/2} Z_i$. Then, the projection is the same considering inputs $\tilde{X}_i' = \Sigma_H^{1/2} \frac{Z_i}{\|Z_i\|} \|N_i\|$ where $N_i \sim \mathcal{N}(0, I_d)$. We remark that under Assumption 4, we have $\frac{Z_i}{\|Z_i\|} \|N_i\| \sim \mathcal{N}(0, I_d)$, then $\tilde{X}_i'$ is a Gaussian vector. In consequence, without loss of generality, we assume that $\tilde{X}_i$ is a Gaussian vector for the rest of this section.

## D.2  UPPER-BOUND OF THEOREM 4.10

The upper bound is an application of Theorem 4.7 with $L_H = 3\mathrm{Tr}(\Sigma_H)$ because Assumption 3 is satisfied with $\kappa = 3$ for Gaussian inputs.

## D.3  LOWER-BOUND OF THEOREM 4.10

Let consider $k = \arg\max_{k > n+1}(k-1)(k-n-2)(\mathrm{Tr}(\Sigma_{2:k}^\dagger))^{-1}$.

**Step 1: Decomposition of the noiseless error**  The SVD of $\Sigma_H$ is

$$\Sigma_H = \sum_{j \in [d]} \lambda_j v_j v_j^\top.$$

Using the matrix form of the noiseless error, we have

$$\mathcal{E}(\Sigma_H; 0) = \mathbb{E}\mathrm{Tr}(\Sigma_H(I - P_n)) = \sum_{j \in [d]} \lambda_j \mathbb{E}\mathrm{Tr}(v_j v_j^\top (I - P_n)),$$

where $P_n$ is the orthogonal projection on $(\tilde{X}_i)_{i \in [n]}$. Denoting by $\mathcal{E}_j = \mathbb{E}\mathrm{Tr}(v_j v_j^\top (I - P_n))$, we have

$$\mathcal{E}(\Sigma_H; 0) = \sum_{j \in [d]} \lambda_j \mathcal{E}_j,$$

**Step 2: matrix form of $\mathcal{E}_j$**   Using Lemma H.1 (in particular (47)), we have

$$\mathcal{E}_j = \mathbb{E} \inf_{l \in \mathbb{R}^n} \left\{ \left\| v_j - \sum_i l_i \tilde{X}_i \right\|_2^2 \right\}.$$

We denote by $A_i = (v_j^\top \tilde{X}_i) v_j$, $C_i = \sum_{l=d-k+1}^d (v_l^\top \tilde{X}_i) v_l \mathbf{1}_{l \neq j}$ and $B_i = \tilde{X}_i - A_i - C_i$. We have $\tilde{X}_i = A_i + B_i + C_i$. By definition $B_i, C_i$ is orthogonal with $v_j$ and $A_i$, and $B_i, C_i$ are orthogonal. Then

$$\mathcal{E}_j = \mathbb{E} \inf_{l \in \mathbb{R}^n} \left\{ \left\| v_j - \sum_i l_i A_i \right\|_2^2 + \left\| \sum_i l_i B_i \right\|_2^2 + \left\| \sum_i l_i C_i \right\|_2^2 \right\}.$$

Thus,

$$\mathcal{E}_j \geq \mathbb{E} \inf_{l \in \mathbb{R}^n} \left\{ \left\| v_j - \sum_i l_i A_i \right\|_2^2 + \left\| \sum_i l_i B_i \right\|_2^2 \right\}.$$

Denoting by $G$ the Gram matrix of $(B_i)_{i \in [n]}$, that is, for all $k, i \in [n]$,

$$G_{ik} = B_i^\top B_k$$

then,

$$\left\| \sum_i l_i B_i \right\|_2^2 = \|l\|_G^2 = \left\| G^{1/2} l \right\|_2^2.$$

Denoting by $\mathbf{A}$ the matrix with columns equal to $(A_1, \ldots, A_n)$ then we have

$$\mathcal{E}_j \geq \mathbb{E} \inf_{l \in \mathbb{R}^n} \left\{ \|v_j - \mathbf{A}l\|_2^2 + \left\| G^{1/2} l \right\|_2^2 \right\}.$$

Furthermore, denoting by $\Sigma^{(j)} = \sum_{l=1}^k \mathbf{1}_{l \neq j} \lambda_l v_l v_l^\top$ then $B_j \sim \mathcal{N}(0, \Sigma^{(j)})$. Remarking that $\mathrm{rank}(\Sigma^{(j)}) \geq k - 1 > n$ then $G$ is almost-surely invertible. In consequence,

$$\mathcal{E}_j \geq \mathbb{E} \inf_{l \in \mathbb{R}^n} \left\{ \left\| v_j - \mathbf{A} G^{-1/2} l \right\|_2^2 + \|l\|_2^2 \right\}.$$

Using Lemma H.1, we have

$$\mathcal{E}_j \leq \mathbb{E} \mathrm{Tr} \left( v_j v_j^\top (\mathbf{A} G^{-1} \mathbf{A}^\top + I)^{-1} \right).$$

**Step 3: Fubini and Jensen theorems**   $\mathbf{A}$ and $G$ are independent because $(A_i)$ and $(B_i)$ are independent, then

$$\mathcal{E}_j \geq \mathbb{E} \mathbb{E} \left[ \mathrm{Tr} \left( v_j v_j^\top (\mathbf{A} G^{-1} \mathbf{A}^\top + I)^{-1} \right) | A \right].$$

By convexity of inverse operator,

$$\mathbb{E} \left[ \mathrm{Tr} \left( v_j v_j^\top (\mathbf{A} G^{-1} \mathbf{A}^\top + I)^{-1} \right) | A \right] \geq \mathrm{Tr} \left( v_j v_j^\top (\mathbf{A} \mathbb{E}[G^{-1} | A] \mathbf{A}^\top + I)^{-1} \right).$$

Using Corollary H.4, $\mathbb{E}[G^{-1} | A] = \mathbb{E}[G^{-1}] = \sigma_j^{-2} I$ with $\sigma_j^{-2} := \mathbb{E} \mathrm{Tr}(G^{-1})/n$. Then

$$\begin{aligned}
\mathcal{E}_j &\geq \mathbb{E} \mathrm{Tr} \left( v_j v_j^\top (\mathbf{A} (1/\sigma_j^2) I_n \mathbf{A}^\top + I)^{-1} \right) \\
&= \sigma_j^2 \mathbb{E} \mathrm{Tr} \left( v_j v_j^\top (\mathbf{A} \mathbf{A}^\top + \sigma_j^2 I)^{-1} \right) \\
&\geq \sigma_j^2 \mathrm{Tr} \left( v_j v_j^\top (\mathbb{E}[\mathbf{A} \mathbf{A}^\top] + \sigma_j^2 I)^{-1} \right).
\end{aligned}$$

Futhermore, $\mathbb{E}[\mathbf{A} \mathbf{A}^\top] = n \lambda_i v_i v_i^\top$ then

$$\mathcal{E}_j \geq \frac{\sigma_j^2}{n} \frac{1}{\lambda_j + \sigma_j^2/n}. \tag{39}$$

**Step 4: $\sigma_j^2$ lower bound**  Using Corollary H.4,

$$\sigma_j^2 = \frac{n}{\mathbb{E}\mathrm{Tr}(G^{-1})}$$
$$\geq (k-1)(k-n-2)(\mathrm{Tr}((\Sigma^{(j)})^{-1})^{-1}$$
$$\geq \sigma_0^2$$

**Step 5: putting things together**  Combining the previous steps gives

$$\mathcal{E}(\Sigma_H;0) = \sum_{j\in[d]} \lambda_j \mathcal{E}_j$$
$$\geq \sum_{j\in[d]} \lambda_j \frac{\sigma_j^2}{n} \frac{1}{\lambda_j + \sigma_j^2/n}$$
$$\geq \sum_{j\in[d]} \lambda_j \frac{\sigma_0^2}{n} \frac{1}{\lambda_j + \sigma_0^2/n}$$
$$= \frac{\sigma_0^2}{n} \mathrm{df}_1(\Sigma_H, \sigma_0^2/n),$$

with $\sigma_0^2 = \max_{k>n+1}(k-1)(k-n-2)(\mathrm{Tr}(\Sigma_{2:k}^{\dagger}))^{-1}$.

### D.4 Example of $\sigma_0^2$ lower bounds

- Isotropic case: where $\lambda_1 = \cdots = \lambda_d$,

$$\sigma_0^2 \geq \max_{k>n+1}(k-1)(k-n-2)(\mathrm{Tr}(\Sigma_{2:k}^{\dagger}))^{-1}$$
$$\geq \max_{k>n+1}(k-n-2)\lambda_1$$
$$\geq (d-n-2)\lambda_1$$
$$= \left(1 - \frac{n+2}{d}\right)\mathrm{Tr}(\Sigma_H).$$

- Large minimum eigenvalue (near isotropic case):

$$\sigma_0^2 \geq (d-n-2)\lambda_d.$$

- Comparison with $\lambda_n$:

$$\sigma_0^2 \geq \max_{k>n+1}(k-1)(k-n-2)(\mathrm{Tr}(\Sigma_{2:k}^{\dagger}))^{-1}$$
$$\geq \max_{k>n+1}(k-n-2)\lambda_k$$
$$\geq \lambda_{n+3}.$$

- Specific cases: $\lambda_j = 1/j$ then $\mathrm{Tr}(\Sigma_H) \sim \log(d)$ and

$$\sigma_0^2 \geq \max_{k>n+1}(k-1)(k-n-2)(\mathrm{Tr}(\Sigma_{2:k}^{\dagger}))^{-1}$$
$$= \max_{k>n+1} 2\frac{k-n-2}{k}$$
$$= 2(1-(n+2)/d).$$

The difference is a factor log.

## D.5 OPTIMALITY OF THE LOWER/UPPER BOUNDS AND PROOF OF COROLLARY 4.14

The aim of this section is to prove that the two bounds are close to a constant factor in high dimensions. We can start with the following computation:

$$
\begin{aligned}
\frac{\bar{\lambda}_0 \mathrm{df}_1(\Sigma_H; \bar{\lambda}_0)}{\underline{\lambda}_0 \mathrm{df}_1(\Sigma_H; \underline{\lambda}_0)} &\leq \frac{\bar{\lambda}_0}{\underline{\lambda}_0} \\
&= \frac{3\mathrm{Tr}(\Sigma_H)}{\sigma_0^2} \\
&\leq \frac{3}{1 - \frac{n-2}{d}} \frac{\mathrm{Tr}(\Sigma_H)}{d} \frac{\mathrm{Tr}(\Sigma_{H,2:d}^{-1})}{d-1}.
\end{aligned}
$$

*Proof of Corollary 4.14.* Assume that $\lambda_j = j^{-\alpha}$. First, if $1 > \alpha > 0$, we have,

$$
\begin{aligned}
\mathrm{Tr}(\Sigma_H) &= \sum_{j=1}^{d} \frac{1}{j^{\alpha}} \\
&\leq 1 + \int_1^d x^{-\alpha} dx \\
&= 1 + \frac{d^{1-\alpha} - 1}{1 - \alpha} \\
&\leq \frac{-\alpha}{1 - \alpha} + \frac{d^{1-\alpha}}{1 - \alpha}.
\end{aligned}
$$

And,

$$
\begin{aligned}
\mathrm{Tr}(\Sigma_{H,2:d}^{\dagger}) &= \sum_{j=2}^{d} j^{\alpha} \\
&\leq \int_2^{d+1} x^{\alpha} dx \\
&= \frac{(d+1)^{1+\alpha} - 2^{1+\alpha}}{1 + \alpha} \\
&\leq \frac{(d+1)^{1+\alpha}}{1 + \alpha}.
\end{aligned}
$$

Thus,

$$
\mathrm{Tr}(\Sigma_H)\mathrm{Tr}(\Sigma_{H,2:d}^{\dagger}) \leq \frac{1}{(1+\alpha)(1-\alpha)}(-\alpha(d+1)^{1+\alpha} + (d+1)^{1+\alpha}d^{1-\alpha}).
$$

Then, using $\alpha < 1$,

$$
\frac{\mathrm{Tr}(\Sigma_H)}{d} \frac{\mathrm{Tr}(\Sigma_{H,2:d}^{-1})}{d-1} \leq \frac{1}{(1+\alpha)(1-\alpha)} \left(\frac{d+1}{d-1}\right)^{\alpha}.
$$

Then, for $d > 3$,

$$
\frac{\bar{\lambda}_0 \mathrm{df}_1(\Sigma_H; \bar{\lambda}_0)}{\underline{\lambda}_0 \mathrm{df}_1(\Sigma_H; \underline{\lambda}_0)} \leq \frac{3}{1 - \frac{n-2}{d}} \frac{2^{1+\alpha}}{(1+\alpha)(1-\alpha)}. \tag{40}
$$

Then, if $\alpha = 1$, using similar arguments, we have

$$
\begin{aligned}
\mathrm{Tr}(\Sigma) &= \sum_{j=1}^{d} \frac{1}{j^{\alpha}} \\
&\leq 1 + \int_1^d x^{-1} dx \\
&= 1 + \log(d),
\end{aligned}
$$

and

$$\text{Tr}(\Sigma_{2:d}^{\dagger}) = \sum_{j=2}^{d} j$$

$$= \frac{d(d+1) - 2}{2}$$

$$\leq \frac{d(d+1)}{2}.$$

We obtain, for $d > 3$,

$$\frac{\bar{\lambda}_0 \text{df}_1(\Sigma_H; \bar{\lambda}_0)}{\underline{\lambda}_0 \text{df}_1(\Sigma_H; \underline{\lambda}_0)} \leq \frac{6}{1 - \frac{n-2}{d}} \log(d). \tag{41}$$

$\square$

## D.6 PROOF OF THEOREM 4.15

**Lemma D.1.** *If $X$ has a centered gaussian distribution with $\Sigma \succ 0$ and $n < d - 1$, then*

$$\bar{\mathcal{E}}(\Sigma, \sigma^2) \leq \frac{\sigma^2 d}{n - d - 1}.$$

*Proof.* We use the variational form with $l_i(X) = X_i^{\top} \hat{\Sigma}^{-1} X$, the bias is zero for this choice, thus

$$\bar{\mathcal{E}}(\Sigma, \sigma^2) \leq \sigma^2 \mathbb{E} \sum_{i \in [n]} l_i^2(X)$$

$$= \frac{\sigma^2}{n} \mathbb{E} \text{Tr}(\Sigma \hat{\Sigma}^{-1})$$

$$= \frac{\sigma^2}{n} \mathbb{E} \text{Tr}((\Sigma^{-1/2} \hat{\Sigma} \Sigma^{-1/2})^{-1})$$

$$= \sigma^2 \mathbb{E} \text{Tr}(W^{-1}),$$

with $W \sim \mathcal{W}_n(I_d)$. Thus

$$\bar{\mathcal{E}}(\Sigma, \sigma^2) \leq \frac{\sigma^2 d}{n - d - 1}.$$

$\square$

**Proposition D.2.** *If inputs are gaussian, we have for two non-negative matrix $A$ and $B$,*

$$\bar{\mathcal{E}}(A + B; 0) \leq \bar{\mathcal{E}}(A; \text{Tr}(B)) + \text{Tr}(B).$$

*Proof.* Let start by the decomposition, $X_i = X_i^A + X_i^B$ with $X_i^A \sim \mathcal{N}(0, A)$ and $X_i^B \sim \mathcal{N}(0, B)$.

$$\bar{\mathcal{E}}(A + B; 0) = \mathbb{E} \inf_{l \in \mathbb{R}^n} \left\| X^A + X^B - \sum l_i (X_i^A + X_i^B) \right\|_2^2$$

Thus,

$$\bar{\mathcal{E}}(A + B; 0) = \mathbb{E} \inf_{l \in \mathbb{R}^n}$$

$$\left\{ \left\| X^A - \sum l_i X_i^A \right\|_2^2 + \left\| X^B - \sum l_i X_i^B \right\|_2^2 + 2 \left( X^A - \sum l_i X_i^A \right)^{\top} \left( X^B - \sum l_i X_i^B \right) \right\}$$

Using tower rules (marginalizing over $X_i^B$ and $X^B$), and inequality $\mathbb{E} \inf \leq \inf \mathbb{E}$, we found

$$\bar{\mathcal{E}}(A + B; 0) \leq \mathbb{E} \inf_{l \in \mathbb{R}^n} \left\{ \left\| X^A - \sum l_i X_i^A \right\|_2^2 + \mathbb{E} \left[ \left\| X^B - \sum l_i X_i^B \right\|_2^2 \right] \right\}.$$

Furthermore, $(X_i^B), X^B$ are i.i.d. and centered thus

$$\mathbb{E}\left[\left\|X^B - \sum l_i X_i^B\right\|_2^2\right] = \mathbb{E}[\|X^B\|_2^2] + \sum_i l_i^2 \mathbb{E}[\|X_i^B\|_2^2]$$

$$= \mathrm{Tr}(B) + \mathrm{Tr}(B)\sum_i l_i^2.$$

Then,

$$\bar{\mathcal{E}}(A+B;0) \leq \mathbb{E}\inf_{l\in\mathbb{R}^n}\left\{\left\|X^A - \sum l_i X_i^A\right\|_2^2 + \mathrm{Tr}(B)\sum_i l_i^2\right\} + \mathrm{Tr}(B)$$

$$= \bar{\mathcal{E}}(A;\mathrm{Tr}(B)) + \mathrm{Tr}(B).$$

$\square$

*Proof of Theorem 4.15 upper-bound.* Let $k < n-1$. Let the SVD $\Sigma_H = \sum_{j=1}^d \lambda_j v_j v_j^\top$. We used the previous lemma for $A = \sum_{j=1}^k \lambda_j v_j v_j^\top$ and $B = \sum_{j=k+1}^d \lambda_j v_j v_j^\top$. We have

$$\bar{\mathcal{E}}(\Sigma_H;0) \leq \bar{\mathcal{E}}(A;\mathrm{Tr}(B)) + \mathrm{Tr}(B).$$

Using Lemma D.1,

$$\bar{\mathcal{E}}(\Sigma_H;0) \leq \mathrm{Tr}(B)\frac{k}{n-k-1} + \mathrm{Tr}(B).$$

Using $\mathrm{Tr}(B) = R_k$, we conclude

$$\bar{\mathcal{E}}(\Sigma_H;0) \leq R_k \frac{n-1}{n-k-1}.$$

$\square$

*Proof of Theorem 4.15 lower-bound.* We have $\bar{\mathcal{E}}(\Sigma_H;0) = \mathbb{E}\mathrm{Tr}(\Sigma_H(I-P))$ where $P$ is the orthogonal projection on $(\tilde{X}_i)_{i\in[n]}$. Using Von Neumann's trace inequality, we have

$$\mathrm{Tr}(\Sigma_H P) \leq \sum_{j\in[d]} \lambda_j(\Sigma_H)\lambda_j(P)$$

$$= \sum_{j\in[n]} \lambda_j(\Sigma_H),$$

because, as an orthogonal projection on $n$ observations, $\lambda_j(P) = 1$ for $j \leq n$ and $0$ for $j > n$. Then

$$\mathrm{Tr}(\Sigma_H(I-P)) = \mathrm{Tr}(\Sigma_H) - \mathrm{Tr}(\Sigma_H P) \geq \sum_{j>n} \lambda_j(\Sigma_H).$$

Furthermore, $\sum_{j>n} \lambda_j(\Sigma_H) = R_n$, thus

$$\bar{\mathcal{E}}(\Sigma_H;0) = \mathbb{E}\mathrm{Tr}(\Sigma_H(I-P)) \geq R_n.$$

$\square$

# E    PROOFS FOR EXAMPLE 4.3

**Lemma E.1.** *If $\lambda_j(\Sigma) = j^{-\alpha}$ for $\alpha > 1$, then for all $\lambda > 0$*

$$\mathrm{df}_1(\Sigma, \lambda) \leq C_\alpha \lambda^{1/\alpha}.$$

*Proof.*

$$\mathrm{df}_1(\Sigma, \lambda) = \sum_{j=1}^{d} \frac{j^{-\alpha}}{j^{-\alpha} + \lambda}$$

$$= \sum_{j=1}^{d} \frac{1}{1 + \lambda j^{\alpha}}$$

$$\leq \int_{0}^{+\infty} \frac{1}{1 + x^{\alpha}\lambda} dx,$$

because $\alpha > 1$. Using $y = \lambda x^{\alpha}$, $x = y\lambda^{1/\alpha}$ thus

$$\mathrm{df}_1 f(\Sigma, \lambda) \leq \lambda^{1/\alpha} \int_{0}^{+\infty} \frac{1}{1 + y^{\alpha}} dy.$$

We conclude using $C_\alpha = \int_{0}^{+\infty} \frac{1}{1+y^{\alpha}} dy < +\infty$. □

*Proof of Example 4.3.* $\Sigma_H = \rho^2 \Sigma^{2r}/\mathrm{Tr}(\Sigma^{2r})$ then, using Theorem 4.1,

$$\bar{\mathcal{E}}(\Sigma_H; \sigma^2) \leq \frac{\sigma^2 + 3\mathrm{Tr}(\Sigma_H)}{n} \mathrm{df}_1 \left(\Sigma_H, \frac{\sigma^2 + \kappa\mathrm{Tr}(\Sigma_H)}{n}\right)$$

$$= \frac{\sigma^2 + \kappa\rho^2}{n} \mathrm{df} \left(\Sigma^{2r}, \frac{\sigma^2/(\rho^2\mathrm{Tr}(\Sigma^{2r})) + \kappa}{n}\right)$$

$$\leq \frac{\sigma^2 + \kappa\rho^2}{n} \mathrm{df}_1 \left(\Sigma^{2r}, \frac{\sigma^2/\rho^2 + \kappa}{n}\right),$$

using $\mathrm{Tr}(\Sigma^{2r}) \geq 1$. Then, using Lemma E.1, for $\lambda_j(\Sigma^{2r}) = j^{-2\alpha r}$, we have

$$\bar{\mathcal{E}}(\Sigma_H; \sigma^2) \leq C_{2\alpha r} \frac{\sigma^2 + \kappa\rho^2}{n} \left(\frac{\sigma^2/\rho^2 + \kappa}{n}\right)^{1/2\alpha r}$$

$$\leq C_{2\alpha r}\rho^2 \left(\frac{\sigma^2/\rho^2 + \kappa}{n}\right)^{1-1/2\alpha r}.$$

□

**Lemma E.2.** *Let $S_{m,p} = \sum_{j=m}^{p} j^{-\alpha}$ with $0 \leq \alpha \neq 1$, with $p \geq m > 1$, then*

$$\frac{(p+1)^{1-\alpha} - m^{1-\alpha}}{1 - \alpha} \leq S_{m,p} \leq \frac{p^{1-\alpha} - (m-1)^{1-\alpha}}{1 - \alpha}.$$

*In particular,*

- *If $\alpha < 1$,*

$$\frac{(p+1)^{1-\alpha} - m^{1-\alpha}}{1 - \alpha} \leq S_{m,p} \leq \frac{p^{1-\alpha}}{1 - \alpha}.$$

- *If $\alpha > 1$,*

$$\frac{m^{1-\alpha} - (p+1)^{1-\alpha}}{\alpha - 1} \leq S_{m,p} \leq \frac{(m-1)^{1-\alpha}}{\alpha - 1}.$$

*Proof.* Using that $x \longmapsto x^{-\alpha}$ non increasing, we have

$$S_{m,p} = \sum_{j=m}^{p} \frac{1}{j^\alpha}$$

$$\leq \sum_{j=m}^{p} \int_{j-1}^{j} x^{-\alpha} dx$$

$$= \int_{m-1}^{p} x^{-\alpha} dx$$

$$= \frac{p^{1-\alpha} - (m-1)^{1-\alpha}}{1-\alpha}.$$

Using similar arguments,

$$S_{m,p} = \sum_{j=m}^{p} \frac{1}{j^\alpha}$$

$$\geq \sum_{j=m}^{p} \int_{j}^{j+1} x^{-\alpha} dx$$

$$= \int_{m}^{p+1} x^{-\alpha} dx$$

$$= \frac{(p+1)^{1-\alpha} - m^{1-\alpha}}{1-\alpha}.$$

$\square$

Using this lemma, if $\lambda_j(\Sigma) = j^{-\alpha}$ then $\mathrm{Tr}(\Sigma^{2r\alpha})$ is a convergent serie (in $d$) as soon as $2r\alpha > 1$. Thus, there exists $c, C > 0$, that does not depend on $d$, such that $0 < c \leq \mathrm{Tr}(\Sigma^{2r\alpha}) \leq C$. Thus the eigenvalues of $\Sigma_H = \rho^2 \Sigma^{2r}/\mathrm{Tr}(\Sigma^{2r})$ satisfy $C^{-1}\rho^2 j^{-2\alpha r} \leq \lambda_j(\Sigma_H) \leq c^{-1}\rho^2 j^{-2\alpha r}$. Using previous lemma, we obtain

$$C^{-1}\rho^2 \frac{n^{1-2r\alpha} - (d+1)^{1-2r\alpha}}{2r\alpha - 1} \leq R_n \leq c^{-1}\rho^2 \frac{(n-1)^{1-2r\alpha}}{2r\alpha - 1}. \tag{42}$$

## F   PROOF OF SECTION 5

**Lemma F.1.** *If $(X^\top v_j)_{j\in[d]}$ are independent and have symmetric components then for all $R = \sum_{j\in[d]} \epsilon_j v_j v_j^\top$ with $\epsilon \in \{-1,1\}^d$ $RX$ have the same law than $X$.*

*Proof.* $RX = \sum_{j\in[d]} (\epsilon_j v_j^\top X) v_j$. Using that $\epsilon_j v_j^\top X$ has the same law than $v_j^\top X$ and $(X^\top v_j)_{j\in[d]}$ independent, we have $RX$ that have the same law than $\sum_{j\in[d]} (v_j^\top X) v_j = X$ because $(v_j)$ is an orthogonal basis of $\mathbb{R}^d$. $\square$

*Proof of Proposition 5.1.* Let start by recall Lemma B.1:

$$\mathbb{E}[(Y - f(X))^2 | X, (X_i)] = \sigma^2 + \left(\left(X - \sum_{i=1}^{n} l_i((X_i)_i, X)X_i\right)^\top \theta_\star\right)^2 + \sigma^2 \sum_{i=1}^{n} l_i((X_i)_i, X)^2.$$

Let $R = \sum_{j\in[d]} \epsilon_j v_j v_j^\top$, with $\epsilon \in \{-1,1\}^d$, we have $R^{-1} = R^\top$ (orthogonal matrix). Thus

$$= \sigma^2 + \left(\left(RX - \sum_{i=1}^{n} l_i((X_i)_i, X)RX_i\right)^\top R\theta_\star\right)^2 + \sigma^2 \sum_{i=1}^{n} l_i((X_i)_i, X)^2$$

$$= \sigma^2 + \left(\left(RX - \sum_{i=1}^{n} l_i((RX_i)_i, RX)RX_i\right)^\top R\theta_\star\right)^2 + \sigma^2 \sum_{i=1}^{n} l_i((RX_i)_i, RX)^2.$$

because under Assumption 5, $l_i((X_i)_i, X) = l_i((RX_i)_i, RX)$. Using that $RX$ has the same distribution than $X$, we have $\mathbb{E}_{\theta_\star}[(Y - f(X))^2] = \mathbb{E}_{R\theta_\star}[(Y - f(X))^2]$. Thus, integrated $R\theta_\star$ for $(\epsilon_j)_j$ independent Rademacher, gives us

$$\mathbb{E}_{\theta_\star}[(Y - f(X))^2] - \sigma^2 = \mathbb{E}\mathbb{E}_{R\theta_\star}[(Y - f(X))^2] - \sigma^2 \geq \bar{\mathcal{E}}(\nu, \sigma), \tag{43}$$

where $\nu$ is the distribution of $R\theta_\star$. Furthermore, $H = \mathbb{E}[R\theta_\star(R\theta_\star)^\top] = \sum_{j \in d}(v_j^\top \theta_\star)^2 v_j v_j^\top$, thus $\Sigma_H = \sum_{j \in d} \lambda_j (v_j^\top \theta_\star)^2 v_j v_j^\top = \Sigma_{\theta_\star}$. Then

$$\mathcal{E}_{\sigma^2}(f) \geq \bar{\mathcal{E}}(\Sigma_{\theta_\star}, \sigma).$$

$\square$

# G    Prior results on linear algebra and random matrix

## G.1    Singular values decomposition

We provide here a reminder on singular values decomposition and Moore-Penrose pseudoinverse. We can found these results and more on linear algebra in Giraud (2021, appendix).

**Theorem G.1.** *Any $n \times p$ real-valued matrix of rank $r$ can be decomposed as*

$$A = \sum_{j=1}^{r} \sigma_j u_j v_j^\top,$$

*where*

- $\sigma_1 \geq \cdots \geq \sigma_r > 0$,

- $(\sigma_1, \ldots, \sigma_r)$ *are the nonzero eigenvalues of $A^\top A$ and $AA^\top$, and*

- $(u_1, \ldots, u_r)$ *and* $(v_1, \ldots, v_r)$ *are two orthonormal families of $\mathbb{R}^n$ and $\mathbb{R}^p$, such that $AA^\top u_j = \sigma_j^2 u_j$ and $A^\top A v_j = \sigma_j^2 v_j$.*

*Furthermore, the Moore-Penrose pseudo inverse defined as*

$$A^\dagger = \sum_{j=1}^{r} \sigma_j^{-1} v_j u_j^\top,$$

*satisfied*

1. *$A^\dagger A$ is the orthogonal projector on lines of $A$,*

2. *$AA^\dagger$ is the orthogonal projector on columns of $A$,*

3. *$(AO)^\dagger = O^\top A^\dagger$ for any orthogonal matrix $O$.*

## G.2    Symmetric matrix

**Definitions**

- Mahalanobis norm: For a symmetric matrix $A \in \mathbb{R}^{d \times d}$ and $u \in \mathbb{R}^d$, the Mahalanobis notation is defined by
  $$\|u\|_A^2 := u^\top A u.$$
  $\|\|_A$ is a pseudo-norm if $A$ is positive and a norm if $A$ is positive semi-definite.

- Loewner order: for two matrix $A, B$, $A \preceq B$ if and only if $\|\|_A \leq \|\|_B$.

- Operator monotony: a function $f : \mathbb{R}^{d \times d} \to \mathbb{R}^{d \times d}$ is operator monotone if
  $$A \preceq B \Rightarrow f(A) \preceq f(B).$$

- Operator convexity: a function $f : \mathbb{R}^{d \times d} \to \mathbb{R}^{d \times d}$ is operator convex if for all random matrix, defined on positive symmetric matrix, $M$ such that $\mathbb{E}M$ exists,
  $$f(\mathbb{E}M) \preceq \mathbb{E}f(M).$$

**Prior results**

**Proposition G.2.** *We use in this paper the following prior results*

1. *If $C \succeq 0$ then*
$$A \preceq B \Rightarrow \mathrm{Tr}(AC) \leq \mathrm{Tr}(BC).$$

2. *Function $M \longmapsto M^{-1}$ is operator convex and $M \longmapsto -M^{-1}$ is operator monotone on $M \succ 0$.*

3. *$(A, B) \longmapsto ABA$ is operator convex in $A$ and operator monotone in $B$.*

These prior results are classical, see Carlen (2010) for more precisions.

### G.3 RANDOM MATRIX

**Lemma G.3.** *Let $M \in \mathbb{R}^{p \times p}$ be a random symmetric matrix, such that for all vectors $u, v \in \mathbb{S}^{p-1}$, $\mathrm{Law}(u^\top M u) = \mathrm{Law}(v^\top M v)$. Then,*
$$\mathbb{E}M = \frac{\mathbb{E}\mathrm{Tr}(M)}{p} I_p,$$

*and for all $\beta \in \mathbb{R}^p$,*
$$\mathbb{E}\left[\beta^\top M \beta\right] = \|\beta\|_2^2 \frac{\mathbb{E}\mathrm{Tr}(M)}{p}.$$

*This is in particular satisfied if, for any orthogonal matrix $O$, $OMO^\top$ has the same law as $M$.*

*Proof.* By assumption, for all $u, v \in \mathbb{S}^{d-1}$, $\mathbb{E}u^\top M u = \mathbb{E}v^\top M v$. Thus, there exists $\alpha$ such that, for all $v \in \mathbb{S}^d$, $v^\top \mathbb{E}M v = \mathbb{E}v^\top M v = \alpha$, which entails that $\mathbb{E}M = \alpha I$ by characterization of symmetric matrices. Therefore, $\mathbb{E}\mathrm{Tr}(M) = \mathrm{Tr}(\mathbb{E}M) = p\alpha$, and $\mathbb{E}M = \frac{\mathbb{E}\mathrm{Tr}(M)}{p}I$. Hence, for all $\beta \in \mathbb{R}^p$
$$\mathbb{E}\left[\beta^\top M \beta\right] = \beta^\top \mathbb{E}M \beta = \|\beta\|_2^2 \frac{\mathbb{E}\mathrm{Tr}(M)}{p}.$$

The last point easily follows, see for example Page Jr (1984, Proposition 2.14) for the case of invariant distributions by orthogonal transforms.

$\square$

**Lemma G.4.** *For $\theta \sim \rho \mathcal{U}(\mathbb{S}^{d-1})$, then for all matrix $M \in \mathbb{R}^{d \times d}$,*
$$\mathbb{E}[\|\theta\|_M^2] = \frac{\rho^2}{d}\mathrm{Tr}(M).$$

*Proof.*
$$\begin{aligned}
\mathbb{E}[\|\theta\|_M^2] &= \mathbb{E}[\theta^\top M \theta] \\
&= \mathbb{E}\mathrm{Tr}(\theta^\top M \theta) \\
&= \mathbb{E}\mathrm{Tr}(M\theta\theta^\top) \\
&= \mathrm{Tr}(M\mathbb{E}[\theta\theta^\top]),
\end{aligned}$$

Then, $\mathbb{E}[\theta\theta^\top] = aI$ because $O\theta$ has the same law of $\theta$ for all orthogonal matrix $O$. Futhermore, $\mathrm{Tr}(\theta\theta^\top) = \theta^\top \theta = \rho^2$ then $da = \rho^2$, thus $\mathbb{E}[\theta\theta^\top] = \frac{\rho^2}{d}I$. $\square$

## H TECHNICAL LEMMAS

### H.1 RIDGE

**Lemma H.1.** *For $\mathbf{X} \in \mathbb{R}^{n \times d}$ and $y \in \mathbb{R}^n$, the minimizer of*
$$F(\beta) := \|y - \mathbf{X}\beta\|_2^2 + \lambda\|\beta\|_2^2,$$

*is given by $\beta_\lambda = (\mathbf{X}^\top \mathbf{X} + \lambda I)^{-1} \mathbf{X}^\top y$ and*

$$F(\beta_\lambda) = \|y - Py\|_2^2 + \lambda \sum_{i=1}^r \frac{1}{\sigma_i^2 + \lambda}(y^\top u_i)^2 \tag{44}$$

$$= \lambda \mathrm{Tr}(yy^\top (\mathbf{X}\mathbf{X}^\top + \lambda I_n)^{-1}), \tag{45}$$

*where $P$ is the orthogonal projection on columns of $\mathbf{X}$ and the SVD of $\mathbf{X}$ is $\mathbf{X} = \sum_{i=1}^r \sigma_i u_i v_i^\top$.*

*Proof.* $F$ is a strongly convex function, then the minimizer $\beta_\lambda = (\mathbf{X}^\top \mathbf{X} + \lambda I)^{-1} \mathbf{X}^\top y$ is found considering $\nabla F(\beta_\lambda) = 0$. Using $\mathbf{X} = \sum_{i=1}^r \sigma_i u_i v_i^\top$, we have

$$\beta_\lambda = \sum_{i \in [r]} \frac{\sigma_i}{\sigma_i^2 + \lambda}(u_i^\top y)v_i.$$

Thus

$$X\beta_\lambda = \sum_{i \in [r]} \frac{\sigma_i^2}{\sigma_i^2 + \lambda}(u_i^\top y)u_i.$$

Using that $P$ is the orthogonal projection on $u_1, \ldots, u_r$,

$$y - \mathbf{X}\beta_\lambda = y - Py + Py - \mathbf{X}\beta_\lambda$$

$$= y - Py + \sum_{i \in [r]} \frac{\sigma_i^2 + \lambda - \sigma_i^2}{\sigma_i^2 + \lambda}(u_i^\top y)u_i$$

$$= y - Py + \sum_{i \in [r]} \frac{\lambda}{\sigma_i^2 + \lambda}(u_i^\top y)u_i.$$

Then,

$$\|y - \mathbf{X}\beta_\lambda\|_2^2 = \|y - Py\|_2^2 + \sum_{i \in [r]} \frac{\lambda^2}{(\sigma_i^2 + \lambda)^2}(u_i^\top y)^2.$$

Furthermore,

$$\|\beta_\lambda\|_2^2 = \sum_{i \in [r]} \frac{\sigma_i^2}{(\sigma_i^2 + \lambda)^2}(u_i^\top y)^2.$$

Combining these two terms, we found

$$F(\beta_\lambda) = \|y - Py\|_2^2 + \sum_{i \in [r]} \frac{\lambda^2}{(\sigma_i^2 + \lambda)^2}(u_i^\top y)^2 + \lambda \sum_{i \in [r]} \frac{\sigma_i^2}{(\sigma_i^2 + \lambda)^2}(u_i^\top y)^2$$

$$= \|y - Py\|_2^2 + \sum_{i \in [r]} \frac{\lambda(\sigma_i^2 + \lambda)}{(\sigma_i^2 + \lambda)^2}(u_i^\top y)^2$$

$$= \|y - Py\|_2^2 + \lambda \sum_{i \in [r]} \frac{1}{\sigma_i^2 + \lambda}(u_i^\top y)^2.$$

In the case, where the rank $r < n$, we obtain the second equality completing the bases $u_1, ..., u_r$ by $u_{r+1}, ..., u_n$.

$\square$

As a consequence of this lemma, we will use the useful variational characterization.

$$\inf_\beta \{\|y - \mathbf{X}\beta\|_2^2 + \lambda\|\beta\|_2^2\} = \lambda \mathrm{Tr}(yy^\top (\mathbf{X}\mathbf{X}^\top + \lambda I_n)^{-1}). \tag{46}$$

Note that this result is valid for any proper sized $y$ and $\mathbf{X}$. This result can be supplemented by the case $\lambda \to 0^+$,

$$\inf_\beta \{\|y - \mathbf{X}\beta\|_2^2\} = \mathrm{Tr}(yy^\top (I - P))), \tag{47}$$

with $P$ the orthogonal projection on $\mathbf{X}$.

## H.2 MOORE-PENROSE PSEUDOINVERSE

**Lemma H.2** (Trace inequality). *Let $A \succeq 0$, and $A^-$ a reflexive symmetric pseudoinverse, i.e.*

- $A^- A A^- = A^-$,

- $A A^- A = A$,

- $A^- \succeq 0$,

*Then,*

$$\mathrm{Tr}(A^\dagger) \leq \mathrm{Tr}(A^-).$$

*Proof.* We denote $A = \sum_{j \in [r]} \lambda_j v_j v_j^\top$, and we complete the bases by $(v_{r+1}, \ldots, v_d)$,

$$\mathrm{Tr}(A^-) = \sum_{j \in [d]} v_j^\top A^- v_j$$

$$= \sum_{j \in [r]} v_j^\top A^- v_j + \sum_{j=r+1}^d v_j^\top A^- v_j$$

For $j \leq d$, using $A v_j = \lambda_j v_j$,

$$v_j^\top A^- v_j = \frac{1}{\lambda_j^2} v_j^\top A A^- A v_j$$

$$= \frac{1}{\lambda_j^2} v_j^\top A v_j$$

$$= v_j^\top A^\dagger A A^\dagger v_j$$

$$= v_j^\top A^\dagger v_j,$$

using $A^\dagger v_j = (1/\lambda_j) v_j$. Then

$$\mathrm{Tr}(A^-) = \mathrm{Tr}(A^\dagger) + \sum_{j=r+1}^d v_j^\top A^- v_j.$$

We conclude using $A^- \succeq 0$. $\qquad\square$

This lemma is particularly usefull to control the pseudoinverse of a overparametrized Wishart distribution pseudoinverse. $W \sim \mathcal{W}_n(\Sigma)$ if $W = \sum_{i \in [n]} X_i X_i^\top$ where $(X_i) i \in [n]$ are i.i.d $\mathcal{N}(0; \Sigma)$

**Theorem H.3.** *If $d > n + 1$, and $W \sim \mathcal{W}_n(\Sigma)$, then*

$$\mathbb{E}\mathrm{Tr}(W^\dagger) \leq \frac{n}{d} \frac{\mathrm{Tr}(\Sigma^{-1})}{d - n - 1}.$$

*Proof.* We consider the inverse $A^- = \Sigma^{-1/2}(\Sigma^{-1/2} A \Sigma^{-1/2})^\dagger \Sigma^{-1/2}$ that satisfies assumptions of Lemma H.2, thus

$$\mathbb{E}\mathrm{Tr}(W^\dagger) \leq \mathbb{E}\mathrm{Tr}(W^-)$$

$$= \mathbb{E}\mathrm{Tr}(\Sigma^{-1/2}(\Sigma^{-1/2} W \Sigma^{-1/2})^\dagger \Sigma^{-1/2})$$

$$= \mathrm{Tr}(\Sigma^{-1/2}\mathbb{E}[(\Sigma^{-1/2} W \Sigma^{-1/2})^\dagger]\Sigma^{-1/2}).$$

The matrix $\Sigma^{-1/2} W \Sigma^{-1/2} \sim \mathcal{W}_n(I_d)$, then using (Cook and Forzani, 2011) theorem 2.1, we have $\mathbb{E}[(\Sigma^{-1/2} W \Sigma^{-1/2})^\dagger] = \frac{n}{d(d-n-1)} I_d$, then

$$\mathbb{E}\mathrm{Tr}(W^\dagger) \leq \frac{n}{d(d-n-1)}\mathrm{Tr}(\Sigma^{-1}).$$

$\square$

**Corollary H.4** (Inverse of Gramm matrix). *Let $(X_i)_{i \in [n]}$ i.i.d. copies of $\mathcal{N}(0, \Sigma)$, we denote by $G \in \mathbb{R}^{n \times n}$ the Gramm matrix such that $G_{ij} = X_i^\top X_j$. If $n < d - 1$ then $G$ is invertible with*

$$\mathbb{E}G^{-1} = \frac{\mathbb{E}\text{Tr}(G^{-1})}{n}I_n,$$

*and*

$$\mathbb{E}\text{Tr}(G^{-1}) \le \frac{n}{d(d - n - 1)}\text{Tr}(\Sigma^{-1})$$

*Proof.* Let $v \in \mathbb{S}^{n-1}$, we have

$$v^\top G v = \sum_{i,j} v_i G_{ij} v_j$$

$$= \sum_{i,j} v_i X_i^\top X_j v_j$$

$$= \left(\sum_i v_i X_i\right)^\top \left(\sum_j v_j X_j\right).$$

Using $\|v\|_2 = 1$, we remarks that $\sum_i v_i X_i \sim \mathcal{N}(0, \Sigma)$ thus the law of $v^\top G v$ does not depends on $v$. In other words, for all orthogonal matrix $O$, $OGO^\top$ and $G$ have the same law. Thus, $OG^{-1}O^\top = (O^\top GO)^{-1}$ has the law of $G^{-1}$. Using, Lemma G.3, we have $\mathbb{E}G^{-1} = \frac{\text{Tr}(G^{-1})}{n}I_n$. Furthermore, $G^{-1}$ have the same spectra than $W^\dagger$ with $W = \sum X_i X_i^\top$. We conclude using Theorem H.3. $\square$

