# OpenReview forum: "Breaking the curse of dimensionality for linear rules: optimal predictors over the ellipsoid"
_ICLR.cc/2026/Conference — Submitted to ICLR 2026_

### Official Review · Reviewer_nHwt · 2025-10-30

**Soundness:** 2
**Presentation:** 2
**Contribution:** 3
**Rating:** 6
**Confidence:** 3

**Summary:**

The paper studies bounds on the minimax rate of error for learning linear data with models following a linear prediction rule. In the high-dimensional settings it was previously known that the minimax rate is infinity when ground truth $\theta^*$ is unbounded. The paper considers the restricted setup of $\theta^ *$ from an ellipsoid and derive a lower-bound on the minimax rate for this problem by tightly bounding the "optimal averaged excess risk" which is a lower-bound on the minimax rate. In particular, it follows that the optimal solution for this problem is in the forms of ridge-regression of (adjusted) features and the optimal error connects nicely to the 1th and 2nd-degree of freedom. The results of the paper follows by the analysis of this term.

**Strengths:**

The results and the studied problem are quite well-written and easy to understand. The paper also tackles an interesting theoretical problem on the minimax rate. The result of the first theorem (thm 3.1.) stating that the optimal averaged risk reduces to ridge regression on transformed covariates is quite intuitive.  The results are rigorous and the paper provides sharp bounds for the minimax rate, which can be useful for future works.

**Weaknesses:**

-It's better for clarity if the authors also include the optimal $l^\star$ in the statement of the theorem 3.1.

-The prior works section is also quite compact. I'm not much familiar with prior works on minimax rates of linear regression, therefore I cannot comment on the novelty of results or approach.  I believe a more detailed comparison in terms of technical novelties is lacking in the current version. For example, have the tricks in Eq. (16) or to lower-bound the minimax rate based on the averaged rate used in prior works before? How do the results on fixed $\theta^*$ compare to previous results?

-The implications and impacts of results are vague. Although, the paper can also benefit from numerical experiments, I consider this a minor weakness of the paper.

typos:
Theorem 3.1. is stated as proposition 3.1.

line 243: upper bound->lower bound?

theorem 2.5 in line 314.

**Questions:**

See the section above.

---

> ### Author Response · Authors · 2025-11-25
>
> First, we thank you for your comments and suggestions, which allowed us to produce an improved version of our article. We have incorporated several of your remarks and questions into the revised manuscript. All modifications are visible in the current version of the paper.
>
>
> **W1)** This has been implemented in Eq. 11—thank you for the suggestion.
>
> **W2)** Under the source condition, many asymptotic results already exist (we added a reference in Remark 3.2(a)), including analyses of the mini-averaged risk (e.g., Richard 2021). Our work focuses on *non-asymptotic* guarantees, which are well established in the low-dimensional regime $n > d$ but much less explored in the overparameterized regime $d > n$. The decomposition in eq 16 is a particularity of our work but, as explained, its can be seen as a bias variance decomposition.  The analysis for a fixed $\theta_\star$  under Assumption 5 is, to our knowledge, new; as explained on line 468, we adapt a technique from Richard 2021 to derive this lower bound.
>
> **W3)** We are currently working on expanding the discussion of the implications of our results, and some of these additions already appear in the current version of the paper.
>
>
> Thank you for pointing out the typos; we have corrected most of them.

---

### Official Review · Reviewer_CCAX · 2025-11-01

**Soundness:** 3
**Presentation:** 2
**Contribution:** 2
**Rating:** 2
**Confidence:** 3

**Summary:**

This paper studies how problem structure can mitigate the curse of dimensionality in regression with linear predictors. Bounds are given in terms of notions of effective dimension. The primary quantity of interest is the best average-case excess risk with respect to a prior distribution supported on an ellipsoid. New lower and upper bounds are derived both in noisy and noiseless settings. Several examples are given that instate the bounds for particular input data covariance matrices.

**Strengths:**

The paper is well written and mostly clear to follow from beginning to end. Some of the proofs appear to be novel, which is hard to find in many linear regression papers. In particular, the split into noiseless and noisy errors is interesting and possibly novel; see Eqn 16. Overall my impression is that the paper is original and of fair quality.

**Weaknesses:**

While the paper does succeed in proving several interesting results, these results seem loosely connected. The paper does not tie the result together into a single cohesive takeaway message. Several different assumptions and definitions are used in different sections of the paper.
There are numerous typos and notation issues that hurt the readability of the paper. The paper does not seem to be in publication-ready shape yet.

Moreover, the paper does not sufficiently motivate the mini-averaged risk over the minimax risk formulation. There is some discussion in Eqn 9, but until then the paper makes it seem that the minimax risk is the object of interest given the attention paid to it by prior work. Is the min-averaged risk interesting more fundamentally, beyond just the fact that it can be much smaller than the minimax risk?
The paper would be stronger if the authors could upper bound the constrained minimax risk $ \inf_g\sup_{\theta^\star\in\Theta}\mathcal{E}_{\sigma^2}(g) $ since the current version of the paper lower bounds this quantity.

**Questions:**

Questions:
1.  Prop. 3.1 seems well-known, especially in Bayesian nonparametric regression where $ \nu $ plays the role of the prior distribution. Would the authors agree? For instance, Knapik, van Zanten, and van der Vaart 2011 Annals of Statistics or similar work.
1. Is equality in Eqn 7 line 166 necessary? Typically other work instead uses norm less than or equal to 1.
1. line 247-250: how can the averaged excess risk be nonzero given the equality in Eqn 12? it seems that Eqn 14 equals zero if and only if eqn 12 equals zero. Maybe $\lambda=\sigma^2/n$ plays a role?

Typos and notation:
1. Optimal averaged risk is not a good name for Eqn 8 because Eqn 5 is already called the averaged excess risk. It is ambiguous whether optimal averaged risk is Eqn 8 or eqn 6. I suggest something like optimal mini-averaged excess risk.
1. line 35 should be $ n\gtrsim \epsilon^{-\frac{2+d}{2}} $
1. Line 83: do you assume the non-increasing ordering of the eigenvalues? Should state that if so.
1. the nicefrac in line fractions look strange and can be confusing, like is line 116--120 where it looks like only $ x' $ is divided by $ h $ instead of the difference; maybe just go back to normal frac or tfrac?
1. line 140: missing word: over the training datatset
1. line 176: spelling of infimum
1. Line 176: why say again? This is the first time the authors restrict the infimum to linear predictors instead of all measurable functions of the data
1. line 179: typo, the constrained the
1. Eqn 11: possible typo, why does the $ n+1 $ index appear in the first term?
1. line 243: lower bound not upper bound
1. Prop 3.1 is called a Theorem 3.1 in the appendix
1. typo line 298 $L_H^2$
1. line 314: example 2.5 not theorem 2.5
1. line 339 spelling of negligible

---

> ### Author Response · Authors · 2025-11-25
>
> First, we thank you for your comments and suggestions, which allowed us to produce an improved version of our article. We have incorporated several of your remarks and questions into the revised manuscript. All modifications are visible in the current version of the paper.
>
>
> **W1)** We have corrected the typos and reorganized the numbering of theorems and assumptions to make the paper more publication-ready, and we continue to improve the manuscript.
>
> **W2)** We have rewritten part of Section 2 to place greater emphasis on the mini-averaged risk, which—as you correctly point out—is the main quantity studied in our paper. In particular, we changed the order of presentation by defining the mini-averaged risk (Eq. 7) before introducing the distribution on the ellipsoid (line 176). Moreover, we added Eq. 15, which allows us to derive lower and upper bounds on the minimax risk using its mini-averaged counterpart.
>
> **Q1)** We agree that the optimal predictor is well known in Bayesian analysis; Proposition 3.1 serves as the starting point of our analysis. Its main contribution is to highlight the compact matrix form of the mini-averaged risk, which is particularly convenient for our purposes. Additional references to similar formulations have been added in Remark 3.2(a).
>
> **Q2)** Equality is not required; only the second moment of  $\theta_\star$  is necessary for our analysis. However, to obtain sharp lower bounds, we often consider  $\theta_\star$  supported on the boundary. We hope that the revised presentation of this section is now clearer.
>
> **Q3)** Yes, $\lambda = \sigma^2 / n$  does play a role. In the limit  $\sigma \to 0^+$  in Eq. 13, only the component of $\sigma^2 (\hat \Sigma_H+\sigma^2/n)^{-1}$  projected onto the  $n < d$  observations tends to zero. The remaining part corresponds to the noiseless error $\mathrm{Tr}(\Sigma_H (I - P_n))$  (Eq. 25).
>
> Thank you for pointing out the typos; we have corrected most of them.

---

### Official Review · Reviewer_fdTs · 2025-11-01

**Soundness:** 1
**Presentation:** 2
**Contribution:** 2
**Rating:** 2
**Confidence:** 3

**Summary:**

Considering a linear regression model $Y_i=X_i^{\rm T}\theta+\epsilon_i$ with the parameter vector $\theta$ distributed on an ellipsoid, this work derives finite upper and lower bounds of the best possible expected excess risk taken over the distribution of $\theta$, when the optimal linear prediction rule is applied. These bounds are based on a result which expresses the optimal expected excess risk as a function of the population and empirical covariances of $\tilde X_i=H^{1/2}X_i$ with $H$ being the second moment of $\theta$, the variance of noise $\epsilon$, and the number $n$ of observations. The finite bounds obtained in the paper are in contrast with the diverging minimax risk in the overparametrized regime, known in the literature.

**Strengths:**

- Characterizing the excess risk of linear models in overparameterized regime is a relevant topic, driven by the need of understanding large learning models such as DNNs.

- The assumption that the parameter vector $\theta$ lies on an ellipsoid is reasonable. The framework of linear prediction rules encompasses a rich range of parametric and non-parametric methods.

**Weaknesses:**

- The presentation lacks clarity, with some referencing issues. For instance, Proposition 3.1 is referred as Theorem 3.1, and I could not find Theorem 4.13 or Theorem 2.5, but only Example 4.13 and Example 2.5.

- The positioning with respect to literature is not well discussed. In the introduction, the authors motivated this analysis mostly by pointing out the contrast between the diverging minimax risk in the overparametrized regime and the bounded typical error for high-dimensional linear model. However, little intuition is provided on why the risk is diverging in the former case and bounded in the latter, which would help greatly in understanding and assessing the contributions of this analysis.

- The main messages are vague. The key question stressed in the abstract on the minimal structural assumptions needed to prevent the curse of dimensionality is not clearly answered. The phrase in Lines 472-473 "In conclusion, assumptions about $\theta$ such as those in Theorem 2.5, with r > 0, are necessary in high-dimensional settings." seems to be relevant, although too implicit. The authors also claimed to "redeem the minimax framework in the overparametrized regime" in Line 054, even though the risk characterized in this article is the average excess risk, not the minimax risk.

**Questions:**

- The assumption that $\theta$ is distributed on an ellipsoid appears to be critical to the theoretical results obtained in this article. Is there any condition on how $\theta$ is distributed on the ellipsoid ? Besides the brief mention of "uniform target weights in the ellipsoid", I could not find any formal statement of the distribution of $\theta$.

- In the proof of Proposition 3.1, the expression of the optimal $l_\star$ involves the unknown second moment $H$ of $\theta$, as it is obtained from $\tilde X_i=H^{1/2}$X_i. Could you clarify this point?

- Could you elaborate on the phrase in Lines 472-473  "In conclusion, assumptions about $\theta$ such as those in Theorem 2.5, with r > 0, are necessary in high-dimensional settings." ?

- Could you state in more mathematical terms what you meant when you said in Lines 263-264 that the noiseless error term in (16) is " equal to the averaged bias of an overparametrized ridgeless regression problem, but lower than the bias of other linear predictor rules", and provide justification for it ?

---

> ### Author Response · Authors · 2025-11-25
>
> First, we thank you for your comments and suggestions, which allowed us to produce an improved version of our article. We have incorporated several of your remarks and questions into the revised manuscript. All modifications are visible in the current version of the paper.
>
> **W1)** Done. The issue with *cleveref* is now resolved.
>
> **W2)** The noiseless error is a fraction of the norm of the best predictor, which may be unbounded. It becomes bounded as soon as the distribution of $\theta_\star$ admits a second moment. We added further intuition on this point in line 160.
>
> **W3)** Yes, we actually study the mini-averaged risk. We revised the text to emphasize this more clearly. Under the source condition, we show that taking $r = 0$  leads to a risk close to that of the naïve predictor $\theta_\star = 0$ , which is too high to obtain meaningful statistical guarantees. Note also that these lower bounds remain valid for the minimax risk.
>
> **Q1)** The only assumption is on the second moment $H$  of $\theta_\star $. If $\theta $ is uniformly distributed on an ellipsoid, this induces a condition on  $H$ ; we refer to Examples 2.4 and 2.5 for details. In particular, Example 2.5 corresponds to the source condition commonly used in high-dimensional settings. We also added a remark after Proposition 3.1 to clarify the link between $H$ and the ellipsoid described by $A$.
>
> **Q2)** The closed form of $l_i$ has been added to the main text. Its indeed depends on the unknown matrix $H$. The goal of the paper is to study the performance of this *theoretical* predictor in order to derive lower bounds on the performance of other predictors (e.g., SGD, ridge, etc.). Furthermore, for the case $H$ is proportional to $I$ (i.e. source condition with $r=1/2$) the optimal linear rules is the ridge with a proper penalization parameter (see ex 4.2).
>
> **Q3)** This item presents an example where the components of $\theta_\star$ and the eigenvectors are not aligned. In this situation, the risk is close to that of the naïve predictor  $f(X)=0$ . Therefore, alignment between $\theta_\star$  and the eigenvectors is necessary to obtain good non asymptotic guarantees. In particular, Example 2.5 with $r>0$ ensures that $\theta_\star$  is concentrated on the leading eigenvectors.
>
> **Q4)** This follows directly from the form of the optimal linear predictor, which we have now added to the main text.

---

### Meta-Review · Area_Chair_BR27 · 2026-01-04

**Summary:**

This paper does not appear ready for publication. Reviewers point out that the literature review does not sufficiently relate the paper to previous work, they highlight a lack of clarity of the take-home message as well as confusion around minimax loss vs mini-averaged loss, and an overall lack of polishing of the paper.

**Reviewer Concerns:**

The technical questions raised by the reviewers were answered by the authors.

**Reviewer Scores:**

I imagine that two of the reviewers would have moved potentially from a score of 2 to a score of 3. But this would still be way below the acceptance threshold and not alter the overall decision.

---

### Decision · Program_Chairs · 2026-01-26

Reject